# Continual Slow-and-Fast Adaptation of Latent Neural Dynamics (CoSFan): Meta-Learning What-How & When to Adapt

**Ryan Missel, Linwei Wang** (`{rxm7244,lxwast}@rit.edu`)
Rochester Institute of Technology

## Abstract

An increasing interest in learning to forecast for time-series of high-dimensional observations is the ability to adapt to systems with diverse underlying dynamics. Access to observations that define a stationary distribution of these systems is often unattainable, as the underlying dynamics may change over time. Naively training or retraining models at each shift may lead to catastrophic forgetting about previously-seen systems. We present a new continual meta-learning (CML) framework to realize continual slow-and fast adaptation of latent dynamics (CoS-Fan). We leverage a feed-forward meta-model to infer *what* the current system is and *how* to adapt a latent dynamics function to it, enabling *fast adaptation* to specific dynamics. We then develop novel strategies to automatically detect *when* a shift of data distribution occurs, with which to identify its underlying dynamics and its relation with previously-seen dynamics. In combination with fixed-memory experience replay mechanisms, this enables continual *slow update* of the *what-how* meta-model. Empirical studies demonstrated that both the meta- and continual-learning component was critical for learning to forecast across non-stationary distributions of diverse dynamics systems, and the feed-forward meta-model combined with task-aware/-relational continual learning strategies significantly outperformed existing CML alternatives.

## 1 Introduction

Learning to forecast from time-series observations of high-dimensional data, such as series of images, is becoming importantly crucial in various applications. Among recent advances, one representative approach is to learn the dynamics function governing these observations in an abstracted latent space as a means to forgo the need for direct supervision (and by extension, direct knowledge) on the system's latent variables (Chung et al., 2015; Krishnan et al., 2015; Karl et al., 2017; Yildiz et al., 2019; Fraccaro et al., 2017; Botev et al., 2021). This is often realized as a sequential latent variable model (sLVM) with a latent dynamics function $z_t = f(z_{<t}; \theta)$ and the latent states' emission back to observation space $x_t = g(z_t)$ (Yildiz et al., 2019; Jiang et al., 2023).

Recent advances have moved towards modeling similar-yet-distinct dynamics that arise from heterogeneous systems (Jiang et al., 2023). One approach that has garnered increasing attention involves learning-to-learn a latent dynamics function via meta-models capable of rapid adaptation. The success of these methods rely on two key fundamental assumptions: 1) training samples are available from *all* systems of interest and this distribution does not change at test time, known as a *stationary* task distribution, and 2) each sample has an identifier linking it to a specific system. These assumptions often break down in time-series forecasting, where data samples arrive as a stream over time such that neither a stationary distribution of systems nor the incoming samples' system identifiers is guaranteed, This represents a scenario of *non-stationary* task distributions with unknown task boundaries or identifiers, where traditional meta-learning approaches have limited applicability.

The emerging research area of *continual meta-learning* (CML) is relevant to lifting these restrictions. CML elevates continual learning (CL) – originally designed to manage non-stationary data distributions (Van de Ven & Tolias, 2019) – to meta-learners over non-stationary task distributions (Riemer et al., 2018; Joseph & Gu, 2021). Examples include weight regularization on meta-models

(He et al., 2019) or sample-storing reservoir to approximate task stationarity (Joseph & Gu, 2021). Unfortunately, existing CML works have predominantly focused on supervised image classification (Joseph & Gu, 2021; Riemer et al., 2018; Caccia et al., 2020) or low-dimensional regression tasks (He et al., 2019), leaving open challenges towards its adoption in latent dynamics forecasting. Two potential challenges stand out. First, current CML research largely employs gradient-based meta-learners, such as model-agnostic meta-learning (MAML) (Finn et al., 2017) which, although effective for image-based tasks, can fail to generalize to a broader range of problem domains. This limitation was noted in meta-reinforcement-learning that focuses on short-term forecasting of action sequences (Mishra et al., 2017), Second, many CML approaches lack a task-identification mechanism (Joseph & Gu, 2021; Riemer et al., 2018), which prevents them from fully utilizing the potential of bi-level optimization for the meta-learners to learn to adapt to all previously seen tasks.

**Our contributions.** This work introduces a new *what-how & when* framework for continual slow-and-fast adaptation of latent neural dynamics (*CoSFan*) across sequentially presented non-stationary distributions of dynamics systems (*i.e.*, tasks,) all without prior knowledge of task boundaries or identifiers. It has two major contributions. First, we propose a *what-how* meta-model to infer *what* the current system is and *how* to adapt a dynamics function to it (Jiang et al., 2023), in a feed-forward manner to realize *fast adaptation* to specific dynamics systems. We show that this feed-forward meta-model provides a stronger algorithmic prior than gradient-based optimization (*e.g.*, MAML) for adaptation, offering significantly faster adaptation and improved predictive performance.

Second, we propose novel strategies to automatically detect *when* a shift of data distribution occurs and, based on which, to identify the underlying dynamics systems and their relations. In combination with fixed-memory experience replay mechanisms, this enables pairing of context-query samples to fully utilize bi-level optimization for continual *slow update* of the *what-how* meta-models. Specifically, we propose two alternative strategies: a simple boundary-based pseudo-labeling mechanism, paired with standard reservoir sampling (Vitter, 1985), to maintain a stationary approximation of the observed samples and their tasks IDs over time; and a cluster-based labeling mechanism, employing continual Bayesian Gaussian mixture models (GMM) on the *what*-embedding extracted by the meta-model, to approximate task distribution via a distilled representation of task relations.

We evaluated *CoSFan* in time-series forecasting in two comparative studies. First, we compared *CoSFan* with state-of-the-art latent dynamics models and their continual extensions, evaluating the *necessity* of both the meta- and continual-components in learning to adapt across non-stationary task distributions. Second, we compared with its alternative formulations based on existing CML works, evaluating the *benefits* of the proposed *what-how* meta-models and their continual learning strategies. In a series of increasingly complex continual learning settings with high-dimensional representations of Hamiltonian systems, we quantitatively and qualitatively demonstrated *C*'s capacity to detect and learn new tasks while preventing catastrophic forgetting in sequential tasks. Models and experimental codes are available at `https://github.com/qu-gg/CoSFan`.

## 2 RELATED WORKS

**Learning-to-learn latent dynamics.** Recent advances have focused on modeling similar-yet-distinct dynamics manifested as heterogeneous samples, particularly through meta-learning (Jiang et al., 2023; Wang et al., 2022; Kirchmeyer et al., 2022) and multi-environment learning (Yin et al., 2021; Zintgraf et al., 2019). One effective approach of increasing interest extends latent dynamics functions via meta-learning (Wang et al., 2022; Jiang et al., 2023), where each dynamic system is treated as a task, and meta-models learn-to-learn to rapidly adapt the latent dynamics function to different tasks. Both MAML- (Kirchmeyer et al., 2022) and feed-forward based meta-learners (Jiang et al., 2023) have shown success in real-world applications, such as cross-subject clinical forecasting (Jiang et al., 2022) and cross-buoyancy turbulent flow forecasting (Wang et al., 2022).

Unfortunately, meta-learning fundamentally assumes tasks are independent and identically distributed (Khoee et al., 2024), which can lead to poor generalization for significantly different tasks at meta-test time (Hospedales et al., 2021). Furthermore, it relies on a stationary training distribution, assuming all relevant tasks are present and labeled during training to pair context and query samples for bi-level optimization (Hospedales et al., 2021). In time-series forecasting, these assumptions hinder the continuous aggregation of knowledge as unique dynamic systems may phase in and out

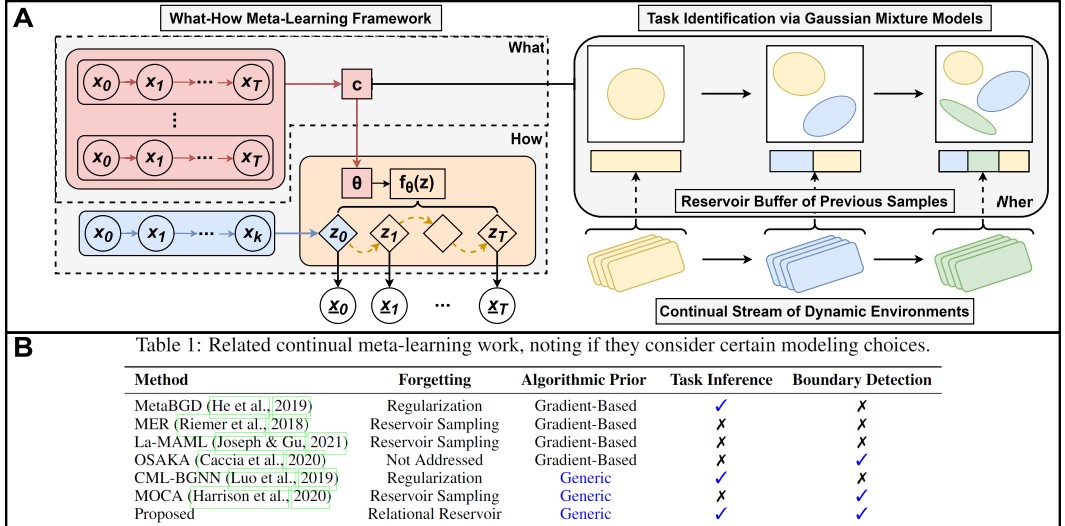

Figure 1: **A)** Overview of *CoSFan*, showing the *what-how* meta-model continually aggregate a heterogeneous data stream with a reservoir that identifies tasks via Gaussian mixture models. **B)** Comparison of the proposed model to relevant CML work.

without prior knowledge. Identifying appropriate auxiliary mechanisms to continually learn how to adapt latent dynamic functions in non-stationary task settings remains an open challenge.

**Continual meta-learning.** Recent developments at the intersection of continual learning and meta-learning, known as *continual meta-learning* (CML), offer inspiration. By leveraging established continual methods, such as reservoir sampling (Riemer et al., 2018; Joseph & Gu, 2021) or Bayesian Gradient Descent (He et al., 2019), on the meta-learner, it allows the meta-learner to be continually trained on a sequence of tasks. However, existing studies focus primarily on supervised image classification tasks (Joseph & Gu, 2021; Riemer et al., 2018; Caccia et al., 2020) or low-dimensional regression problems (He et al., 2019). The potential of CML in the space of time-series or latent dynamics forecasting remains unexplored, leaving two primary open questions.

First, most prior CML works leverage gradient-based meta-learners, specifically MAML (Finn et al., 2017) and its variants. While effective in image and classification domains, MAML's algorithmic prior of gradient descent has been shown to be ill-fit for adapting the task landscape of other domains (Mishra et al., 2017). In meta-reinforcement-learning where the task distributions cannot neatly collapse into one mode, for instance, the shared initialization that MAML uses often results in sub-optimal adaptive performance on isolated tasks (Mishra et al., 2017; Vuorio et al., 2019; Fu et al., 2023). Moreover, gradient-based meta-learners require fine-tuning for every incoming task, *even for known tasks*, which is sensitive to optimization hyper-parameters and computationally inefficient (Nguyen et al., 2021; Mishra et al., 2017). Meta-models with alternative algorithmic priors, such as feed-forward hyper-networks, remains unexplored in the broader scope of CML.

Second, meta-learning's signature bi-level optimization depends on known task identifiers to pair context and query samples. How to achieve this in *task-agnostic* settings – where neither task boundary nor identifier is known – is unresolved. Existing CML approaches bypass this through either regularization techniques (He et al., 2019; Harrison et al., 2020; Caccia et al., 2020) or MAML-based strategies to align gradients between new and previous tasks in a memory buffer (Riemer et al., 2018; Joseph & Gu, 2021). Limitations of the latter are discussed above, while the former often faces capacity saturation and instability in high-dimensional settings (Joseph & Gu, 2021).

*CoSFan* breaks through the reliance of mainstream CML methods on MAML-variants, demonstrating the effectiveness of feed-forward meta-models, combined with automatic task identification and task-relation modeling, in fully leveraging bi-level meta-optimization over non-stationary task distributions. We position our work against prior CML works in Fig. 1B. Discussion of other related areas, *e.g.*, CL and meta-continual-learning, and other algorithmic priors is included in Appendix A.

## 3   PROBLEM FORMULATION

Consider the problem of learning to forecast the trajectory of a data sequence $\mathbf{x}_{0:T}$ when only given access to $l$ initial frames of observations $\mathbf{x}_{0:l}$ ($l \ll T$), where $\mathbf{x}_t \in \mathcal{R}^D$ at time $t$ is high in dimension. Latent dynamics forecasting approaches this by learning the dynamic function governing some low-dimensional latent representations $\mathbf{z}_t \in \mathcal{R}^d$ ($d \ll D$) and its mapping to $\mathbf{x}_t$. We learn to do so from many observed samples of $\mathbf{x}_{0:T}$. Depending on the diversity and distribution of the observed data, below we introduce the setting for this problem in layers, building up from the simplest setting of learning from single systems to learning from non-stationary distributions of multiple systems.

**Single-System Forecasting.**   Let us start with the simplest setting where all observed samples of $\mathbf{x}_{0:T}$ has a shared generative low-dimensional dynamic system $\mathcal{T}$. Leveraging the commonly used sLVM framework (Botev et al., 2021; Jiang et al., 2023), the modeling consists of three components:

*i)* An encoder $\mathbf{z}_0 = enc_\phi(\mathbf{x}_{0:l})$, parameterized by $\phi$, to infer $\mathbf{z}_0$ from the sub-sequence $\mathbf{x}_{0:l}$:

*ii)* A dynamics function $f_\theta$ parameterized by $\theta$, to describe the evolution of $\mathbf{z}_t$ over time and comes in many forms - from linear combinations of transition matrices (Karl et al., 2017; Fraccaro et al., 2017) to neural ordinary differential equations (NODE) (Chen et al., 2018; Yildiz et al., 2019). We opt for the latter in this work, which describes the continuous evolution of $\mathbf{z}_t$ as $\mathbf{z}_t = \mathbf{z}_0 + \int_0^t f_\theta(\mathbf{z}_\tau)d\tau$.

*iii)* An emission function $\mathbf{x}_t = dec_\rho(\mathbf{z}_t)$, parameterized by $\rho$, to map $\mathbf{z}_t$'s to $\mathbf{x}_t$'s in parallel.

Given the final forecasted sequence $\hat{\mathbf{x}}_{0:T}$ from the few-frame input $\mathbf{x}_{0:l}$, the difference in the forecast and the ground truth $\mathcal{L}(\hat{\mathbf{x}}_{0:T}, \mathbf{x}_{0:T})$ can be used as a loss signal to optimize the parameters $\{\phi, \theta, \rho\}$.

**Heterogeneous-Dynamics Extension.**   Now consider an extension where multiple dynamics systems, $\{\mathcal{T}_j\}_{j=1}^M$, exist, each with an unknown system variable $\mathbf{c}(\mathcal{T}_j)$. Examples include varying sets of parameters for a predator-prey model (Kirchmeyer et al., 2022) or image sequences of objects affected by various Hamiltonian systems (Jiang et al., 2023). Under this setting, a fundamental limitation of $f_\theta$ as desdribed above emerges: it learns a *single parameter set* $\theta$ global to all training sequences; as such, it lacks the ability to adapt to unknown dynamics at test-time. It is easy to see then, when only given the partial sequence to forecast from, we need a mechanism to infer *what* the current system is and to know *how* to adapt to this system in order to perform across all dynamics simultaneously. Formally, this means that $f_\theta$ should now be conditioned on a changing context $\mathbf{c}(\mathcal{T}_j)$ as: $\mathbf{z}_t = f_\theta(\mathbf{z}_{<t}, \mathbf{c}(\mathcal{T}_j))$ in order to change forecasting given the same input of past frames.

**Non-Stationary Extension.**   Lastly, consider an additional condition in which the set of dynamics systems, *i.e.,* tasks $\{\mathcal{T}_j\}_{j=1}^M$, are presented in a sequential order $\{\mathcal{T}_1, \mathcal{T}_2, ..., \mathcal{T}_M\}$. We consider the continual-learning scenario where neither the boundary between tasks nor the label of any task is known (Van de Ven & Tolias, 2019). We also assume the presence of *local stationarity* where data from each task are presented for some period of time. This allows the forecasting method to optimize sufficiently and stably before a new task has the potential of appearing.

**Goal of the Learner.**   Overall, we aim to learn a model which can continually adapt the latent dynamics function $\mathbf{z}_t = f_\theta(\mathbf{z}_{<t}; \mathbf{c}(\mathcal{T}_j))$ to different dynamics systems over a non-stationary stream of such systems - without the availability of a known boundary or identifier for these systems.

## 4   METHODOLOGY

We present a *what-how & when* CML framework as illustrated in Fig. 1A, where a fast-adaptation meta-model infers *what* the current task is and *how* to adapt the dynamics function accordingly (Section 4.1), while a slow-adaptation of the meta-model occurs on an approximated stationary task distribution enriched with task identifiers and relations *when* a task shift occurs (Section 4.2).

### 4.1   *What-How* - FAST ADAPTATION BY INFERRING & USING A DYNAMICS' CONTEXT

The concept of *what & how* is broadly applicable across meta-learner algorithms: the *what* component extracts the context $\mathbf{c}(\mathcal{T}_j)$ from $k$ sequences $\{\mathbf{x}_{0:T,j}^{s,i}\}_{i=1}^k = \mathcal{T}_j^s$, called the *context* set for task

$\mathcal{T}_j$; the *how* component maps $\mathbf{c}(\mathcal{T}_j)$ to a task-specific predictive function $f_\theta(\mathbf{z}_{<t}; \mathbf{c}(\mathcal{T}_j))$ to forecast on unknown *query* sequences $\mathbf{x}_{0:T,j}^q$'s. In the commonly used MAML techniques in CML, a shared parameter set $\theta_{init}$ is adapted into a task-specific parameter set $\theta_{task}$ (*what*) by taking a few gradient optimization steps (*how*) over $\mathcal{T}_j^s$. While successful in image classification and low-dimensional regression, they have been shown unsuitable in broader domains, notably in meta-reinforcement learning which act on action sequences of action (Mishra et al., 2017).

We propose a more general *what & how* method for CML that learns the adaptation algorithm in a feed-forward manner, eliminating the need for gradient-based updates at test-time. We use a non-linear encoder, termed the context encoder, to embeds and aggregate the context data into a lower-dimensional representation (*what*), along with a feed-forward hyper-network that transforms this representation into task-specific latent dynamic functions (*how*). We contrast these two what-how models from the lens of algorithmic prior (Section 4.1.2) and adaptation efficiencies (Section 4.1.3).

### 4.1.1 Feed-Forward What-How Meta-Models for Latent Dynamics Forecasting

**What.** Given the *context* set $\mathcal{T}_j^s$, we encode each sequence $\mathbf{x}_{0:T,j}^{s,i}$ to its individual embedding $\mathbf{c}_j^{s,i}$ through the use of a learnable encoding function $e_\psi$. We then use an averaging function to extract the shared knowledge over the set where $k$ is the size of the set:

$$\mathbf{c}(\mathcal{T}_j) = \frac{1}{k} \sum_{i=1}^k \mathbf{c}_j^{s,i}, \text{ where } \mathbf{c}_j^{s,i} = e_\psi(\mathbf{x}_{0:T,j}^{s,i}), \quad \mathbf{x}_{0:T,j}^{s,i} \in \mathcal{T}_j^s. \tag{1}$$

**How.** To use the inferred $\mathbf{c}(\mathcal{T}_j)$ to adapt the latent dynamics function, we employ a *hyper-network* (Ha et al., 2016) $h_\gamma$ to map $\mathbf{c}(\mathcal{T}_j)$ to the task-specific parameters of the latent dynamics function as:

$$\theta = h_\gamma(\mathbf{c}(\mathcal{T}_j)). \tag{2}$$

For the hyper-network architecture, we consider a simple linear transformation as the *what*-encoder already provides a sufficiently complex non-linear mapping to the task context variable.

**Meta-Objective.** The meta-learner's parameters are optimized by minimizing the error between the forecasted and ground-truth *query* sequences across all tasks, with each task evaluated individually in parallel and their errors aggregated. The loss function for a given task is expressed as:

$$\mathcal{L}_{\mathcal{T}_j} = \sum_{x_{0:T,j}^q \sim \mathcal{T}_j} ||\hat{\mathbf{x}}_{0:T,j}^q - \mathbf{x}_{0:T,j}^q||^2, \text{ where } \hat{\mathbf{x}}_{0:T,j}^q = dec_\rho(f_{\theta(\mathbf{c}(\mathcal{T}_j))}(enc_\phi(\mathbf{x}_{0:l,j}^q))). \tag{3}$$

As such, across the tasks, the meta-objective becomes:

$$\min_{\phi,\rho,\gamma,\psi} \sum_{\mathcal{T}_j \sim p(\mathcal{T})} \mathcal{L}_{\mathcal{T}_j}, \tag{4}$$

which jointly optimizes the sLVM's parameters $\{\phi, \rho\}$ and the what-how meta-parameters $\{\gamma, \psi\}$.

### 4.1.2 Hyper-networks as Algorithmic Priors

Recent work in meta-learning has formalized the concept of algorithmic priors, the underlying algorithms used for adaptation, ranging from gradient-based methods to general approaches that learn the adaptation algorithm itself (Mishra et al., 2017). The latter, often realized using recurrent architectures, is sensitive to context set permutations and entangles input representations with the learning algorithm. These factors have historically constrained general meta-learners in complex domains, leading to the prevalent use of gradient-based techniques (Huisman et al., 2023).

In comparison, hyper-networks have seen success in adapting latent dynamics functions in stationary settings (Kirchmeyer et al., 2022). They also have inductive biases that promote weight transfer across tasks while minimizing gradient interference (Jayakumar et al., 2020), which make them a suitable general algorithmic prior for addressing the *transfer-interference trade-off* in non-stationary tasks (Riemer et al., 2018). Meanwhile, unlike gradient-based methods that rely on a global initialization vector for all tasks, feed-forward what-how meta-models learn complex non-linear mappings that structure tasks in an embedding space suited for diverse adaptations. Thus, hyper-networks overcome the limitations of general algorithmic priors while avoiding the constraints of gradient-based methods in diverse task landscapes (Mishra et al., 2017; Vuorio et al., 2019; Fu et al., 2023).

### 4.1.3 Feed-Forward Adaptation Efficiency

Feed-forward approaches are more efficient than gradient-based approaches in both meta-training and meta-testing for two key reasons. First, hyper-networks require only a single forward pass to adapt, avoiding the costly, often repeated, backpropagation steps. Second, gradient-based methods typically require a sequential loop over tasks to aggregate losses and update meta-weights, as most automatic differentiation libraries lack support for parallelized computation branches (Utkarsh et al., 2024). This limits the use of architectures like NODE, where model evaluation involves sequential integration over dynamic functions. In contrast, feed-forward methods can easily parallelize per-task weights (*e.g.*, using vectorized maps) and are unaffected by task quantity.

Dedicated works have resulted in first-order gradient methods for computational efficiency (Nichol et al., 2018) and subsequent CML methods that eliminate the sequential loop (Joseph & Gu, 2021). In Section 5.3, we evaluate these CML approaches in terms of adaptation speed and training times.

### 4.2 *When* - Continual Fast & Slow Adaptation of Latent Dynamics

The learning objective as defined in Equations 3 and 4 is optimized over a stationary distribution of tasks, as in standard meta-learning practice. As such, despite its ability to infer context and adapt its dynamics, its meta-components become susceptible to catastrophic forgetting in non-stationary task settings. Additionally, the absence of task identifiers complicates meta-learning, as pairing context-query data becomes infeasible. Therefore, it is necessary to incorporate mechanisms to automatically detect task boundaries (4.2.1), combat catastrophic forgetting (4.2.2), and identify tasks and their relations to accurately match query samples to their relevant context (4.2.2).

### 4.2.1 Task boundary detection

Similar to many other CML works (Caccia et al., 2020; He et al., 2019), we choose the most recent $k$ observations to function as the context set for the current task. This is a sound choice given the *local stationarity* assumption. When a task shift occurs, the tasks underlying the context and query sequences differ, resulting in a noticeable dip in prediction performance. We leverage this, via a simple yet effective threshold mechanism proposed in (Caccia et al., 2020), to flag a task boundary via $||\mathcal{L}_{j,n} - \mathcal{L}_{j,n-1}|| > \nu$ where $\mathcal{L}_j$ is the current task loss, $n$ is the global batch index, and $\nu$ is determined based on the dataset. Further rationale and details for determining $\nu$ are in Appendix C.7.

### 4.2.2 Reservoir Sampling based on Task IDs and Task Relations

In the non-stationary setting where only the data of one task is actively streaming in at a given time and no task identifiers are available, two challenges arise in applying meta-learning: *i)* how to obtain and aggregate errors from prior tasks to update the meta-weights and *ii)* how to accurately pair context and query samples from the same task, especially for previous tasks without active data.

Existing works approach this by standard reservoir sampling method, a simple algorithm that tracks the number of samples ($N$) seen and, for each incoming sample, overwrites an existing buffer sample with probability $M/N$ where $M$ is the size of the reservoir. To realize meta-learning in this *task-agnostic* reservoir, recent work leverages the approximate equivalence between meta-learning and continual learning objectives in aligning the current task's gradient $\mathcal{T}_j$ with the average gradient of previous tasks $\mathcal{T}_{0:j-1}$ (Riemer et al., 2018; Joseph & Gu, 2021). This alignment can then be achieved by using the current task's samples as context to obtain parameters $\theta_{\mathcal{T}_j}$, and evaluate the meta-loss using $\theta_{\mathcal{T}_j}$ on all past tasks' samples in the reservoir (Joseph & Gu, 2021). We refer to this approach as **Task-Agnostic Reservoir Sampling** and demonstrate in Section 5.3 that this is less effective in CML compared to full bi-level meta-optimization that uses context-query pairing.

**Task-Aware Reservoir Sampling.** To make the reservoir task-aware, we leverage the boundary detection mechanism, assuming each boundary to represent a unique dynamics system and assigning pseudo-labels with a boundary counter. When sampling from the reservoir, context-query pairs are easily matched by using each sample's pseudo-label. In situations where the data stream contains numerous task boundaries or unbalanced task representations, this approach can lead to an increasingly fragmented buffer, resulting in overfitting to specific context-query pairs or quickly forgetting rare tasks under-represented in the buffer. We experimentally validate these issues in Section 5.4.

**Task-Relational Experience Replay.** To make the reservoir be aware of task relations, we leverage a Bayesian Gaussian mixture model (GMM) that automatically identifies clusters within the reservoir over time and determines a sample's relation to them. At detected task boundaries $\mathcal{T}_j \to \mathcal{T}_{j+1}$, we fit the GMM on context-embeddings from the reservoir's samples $C_j^{\text{reservoir}}$ along with an auxiliary memory buffer $C_j^{\text{active}}$ from the active task $\mathcal{T}_j$. To maintain continuity, we reuse the previous GMM components and their weights to initialize the current clustering. New samples are integrated by initializing a new mixture component based on the mean and covariance of $C_j^{\text{active}}$. The weight of this component after refitting determines if these samples represent a new task (high weight) or a re-emerged task (low weight, covered by an existing component). To avoid the number of components from exploding, we remove inactive components with weights below $0.05$.

Given the fit mixture components $\mathcal{Q}$ and the set of context-embeddings $\{C_j^{\text{reservoir}}, C_j^{\text{active}}\}$, the reservoir $\mathcal{R}_j$ is re-balanced in a novel way by allocating $M/\mathcal{Q}_{\text{size}}$ samples to each component, where $\mathcal{Q}_{\text{size}}$ is the number of components. Components with more samples than their allocated size have excess samples discarded, ensuring the reservoir remains within its memory limit.

$$\mathcal{M}_j, \mathcal{R}_j = \sum_{q=1}^{\mathcal{Q}} \pi_q \mathcal{N}(\{C_j^{\text{reservoir}}, C_j^{\text{active}}\}|\mu_q, \sigma_q), \quad \sum_{q=1}^{\mathcal{Q}} \pi_q = 1 \tag{5}$$

where $\pi_q$ is the weight of a component. The resulting sample cluster assignments $\mathcal{M}_j$ serve as pseudo-labels, enabling context-query pairing when sampling the reservoir. In Section 5.4, we show that this approach addresses frequent task transitions and the presence of rare tasks.

### 4.3 Continual Meta-Objective and Optimization

**Continual Approximation of the Meta-Objective.** We can now continually approximate the true stationary meta-objective in Equation 4 as below, bounded by the extent to which the sample reservoir $\mathcal{R}_j$ approximates the true data distribution of a task $\mathcal{T}_j$ and to which the approximated task identifiers $\mathcal{M}_j$ approximate the true task assignments $\mathcal{I}_j$:

$$\min_{\phi,,\rho,\gamma,\psi} \sum_{\mathcal{T}_j \sim p(\mathcal{T})} \mathbb{E}_{(\mathcal{R}_j \sim \mathcal{T}_j, \mathcal{M}_j \sim \mathcal{I}_j)}[\mathcal{L}_{\mathcal{R}_j}(\mathcal{M}_j)]. \tag{6}$$

**Cluster Embedding Regularization.** To guide meta-model optimization to structured clusters, we use regularization from deep clustering (Manduchi et al., 2021) as a MSE between the meta-embeddings $\mathbf{c}_j$ of the reservoir samples and their closest cluster mean $\mu^*$ (by Euclidean distance):

$$\mathcal{L}_{\mathcal{T}_j}^{\text{cluster}} = \sum ||\mathbf{c}(\mathcal{T}_j) - \mu_*||^2 \tag{7}$$

## 5 Experiments

We evaluate *CoSFan* as follows. In Section 5.2, we compared *CoSFan* to representative latent dynamic models with continual extensions, to examine the need of both meta- and continual-formulations in this domain. In Section 5.3, we explored the benefits of *CoSFan*'s what-how and when components over existing CML strategies. In Section 5.4, we analyzed the strengths and limitations of task-aware and task-relational replay strategies. Appendix C contains further ablations.

### 5.1 Experimental Setup

**Data.** First, we considered data where the underlying dynamics is governed by the same equation but with different parameters. We used benchmark image-based time-series of balls bouncing in a box (Fraccaro et al., 2017; Jiang et al., 2023), influenced by $6$ evenly-spaced directions of gravity and sampled $4500$ sequences from each with varying initial positions and velocities. The magnitude of gravitational forces was kept constant. We refer to this as ***gravity-6***. Next, we explored a broader setting involving a collection of different dynamics, each with a range of heterogeneity within the system. In addition to bouncing balls (Fraccaro et al., 2017), we included Hamiltonian pendulums, Hamiltonian mass-spring systems, and two-body systems (Botev et al., 2021). For each dynamics, we selected $3$ parameter configurations, resulting in a total of $12$ dynamics with $1500$ samples each. We refer to this as ***mixed-physics***. Additional data details and visualizations are in Appendix D.0.1.

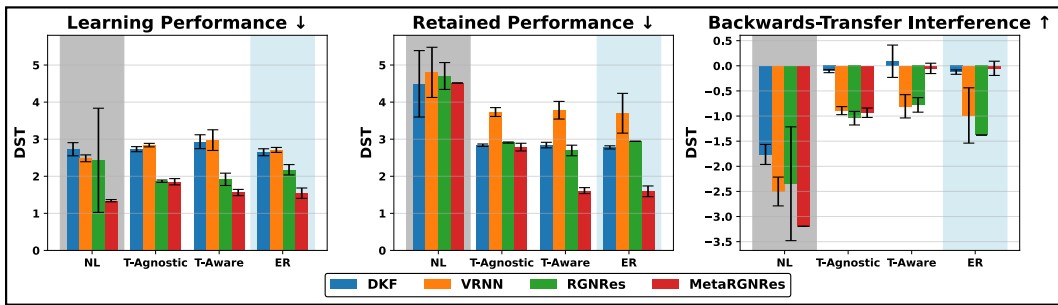

Figure 2: DST performance comparison on **mixed-physics** over existing latent dynamics models and their continual extensions. All methods were run over 5 random seeds.

**Metrics.**    We evaluated two quantitative metrics for latent dynamics forecasting: 1) the MSE of the forecasted images at the pixel level; and 2) a distance metric (DST) proposed by (Jiang et al., 2023) to measure the Euclidean distance between the ground-truth and forecasted objects. To evaluate the ability of a model to learn continually, we followed (Riemer et al., 2018) and presented the above forecasting metrics in two perspectives: *Retained Performance* (RP) and *Learning Performance* (LP). RP metrics represent the average performance across all tasks after they are sequentially considered, emphasizing a model's ability to retain performance on older tasks. LP metrics reflect the average performance on a task immediately after it is learned, measuring a model's effectiveness in incorporating new information. To assess the extent of catastrophic forgetting, we also reported the average difference between LP and RP metrics, referred to as *Backward Transfer and Interference* (BTI), with negative values indicating increased forgetting (Riemer et al., 2018).

When comparing the choice of the what-how meta-models in Section 5.3, we include two additional metrics related to computational efficiency. *Time-to-Adapt-N* (TTA-N) represents the average time it takes, in seconds, for a model to adapt to its context data in the presence of $N$ tasks. *Time-to-Train* (TTT) represents the average time it takes, in minutes, for a model to train over a sequence of tasks.

**Implementation Details.**    Discussion on the implementation-specifics are included in Appendix D.0.3, detailing reservoir sizes, sequence lengths, and hardware considerations.

## 5.2    THE NEED OF WHAT-HOW AND WHEN: COMPARISON WITH EXISTING LATENT DYNAMIC MODELS AND THEIR CONTINUAL EXTENSIONS

**Models.**    We first compared *CoSFan* with representative latent dynamic models and their continual extensions. The baseline latent dynamic models included Deep Kalman Filters (DKF) (Krishnan et al., 2015), Variational Recurrent Neural Networks (VRNN) (Chung et al., 2015), a Residual Recurrent Generative Network (RGNRes) (Botev et al., 2021), and a meta-learning realization of the RGNRes (MetaRGNRes) using the proposed what-how meta-model. We evaluated these four models under four experience replay settings: *i*) Naive Learning (NL) where no past samples are replayed, *ii*) Exact Replay (ER) where the reservoir can accommodate all past samples, *iii*) Task-Agnostic Reservoir Sampling, and *iv*) Task-Aware Reservoir Sampling as described in Section 4.2.2.

**Results.**    We present results on the DST metric on **mixed-physics** in Fig. 2, with **gravity-6** results and complete metrics in Appendix B.1. The gray and blue shaded results respectively represented worst- and best-case scenarios, where the models had no (NL) or full (ER) access to examples from previous tasks. In ER, the addition of the meta-model significantly improved all performance metrics, clearly demonstrating the benefits of the what-how model for learning across dynamic systems.

Looking at LP, the effect of experience replay on a model's ability to learn the "current" task varied across datasets: while performance on **mixed-physics** was relatively unaffected, the absence of experience replay had a greater impact on **gravity-6**. Looking at RP, all models struggled to retain performance on previous tasks when experience replay was absent (NL). The inclusion of a memory component, whether task-agnostic or task-aware as proposed, effectively addressed forgetting and stabilized all models, achieving results comparable to exact replay (ER). However, it is the combination of the what-how meta-model and the task-aware reservoir sampling strategy that delivered

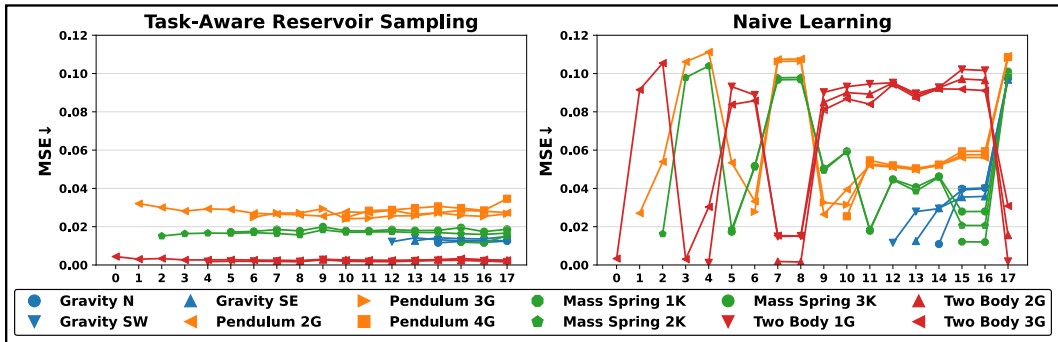

Figure 3: MetaRGNRes with task-aware reservoir (left) overcomes alleviate catastrophic forgetting in the continual setting compared to MetaRGNRes without any experience replay (right).

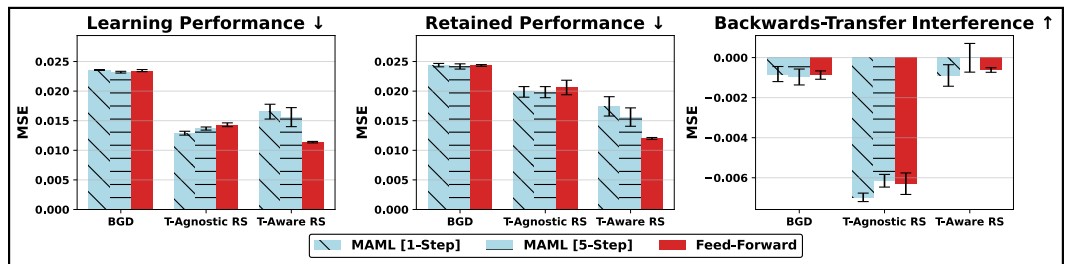

Figure 4: MSE comparison on **gravity-6** over existing CML formulations, averaged over 5 seeds.

significant improvements over all alternatives. This demonstrates the importance of both the continual and meta-learning components, as proposed, for learning to forecast non-stationary dynamics systems. Fig. 3 shows the per-task performance of MetaRGNRes when continually learning over sequentially presented tasks, with (left) and without (right) task-aware experience replay strategies.

## 5.3 DESIGN CHOICES FOR WHAT-HOW & WHEN: COMPARISON WITH EXISTING CML

**Models.** We now considered alternative formulations for each of the what-how and when components in *CoSFan* using existing CML methods (by adopting them from their original implementations). The initial state encoder and decoder backbones, as well as the architecture of the dynamics function, are identical across these baselines. Differences arise from 1) the design of the what-how meta-models, and 2) the continual learning strategy. More specifically:

*Choice of meta-model (what-how):* We considered the proposed feed-forward what-how meta-model in comparison to the MAML-based meta-model most commonly used in existing CML works (Riemer et al., 2018; Joseph & Gu, 2021; Caccia et al., 2020; Harrison et al., 2020; He et al., 2019). We evaluate the MAML method with 1- and 5-inner gradient steps for dynamics adaptation.

*Choice of continual-strategy (when):* We considered the proposed Task-Aware Reservoir Sampling in comparison to two continual-learning strategies used in existing CML works: standard Task-Agnostic Reservoir Sampling from Section 4.2.2 (Riemer et al., 2018; Joseph & Gu, 2021) and uncertainty-based weight regularization via Bayesian Gradient Descent (BGD) (He et al., 2019).

**Results.** Fig. 4 present the MSE results on **gravity-6**, with results on **mixed-physics** and full metrics in Appendix B.2. Among the three continual learning strategies, BGD-based weight regularization (leftmost group) resulted in the lowest performance in both LP and RP. Compared to task-agnostic reservoir, task-aware reservoir showed significant (**gravity-6**) to moderate (**mixed-physics**) improvements in retaining performance and reducing negative interference. While the task-agnostic approach (middle group) showed better LP for MAML (blue) at the cost of catastrophic forgetting, the task-aware approach (rightmost group) showed little negative to even positive transfer.

Comparing meta-model choices, the feed-forward approach with a task-aware reservoir outperformed all other combinations of meta-models and continual strategies. Interestingly, in a task-

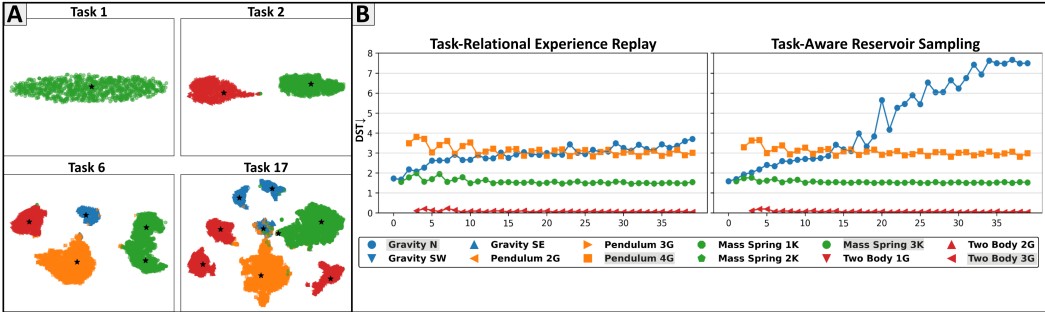

Figure 5: **A**) Task-Relational's reservoir's meta-embeddings throughout **mixed-physics**. Black stars represent GMM mixture means. **B**) Task-Relational vs. Task-Aware strategies on imbalanced tasks where the first task appears once while others cycle. Considered dynamics are highlighted in gray.

agnostic setting, the feed-forward approach performed similarly or worse than MAML. This confirmed our suspicion that while the task-agnostic setting works reasonably well for MAML-based strategies to learn a shared initialization, it limits the potential of general meta-learners. Overcoming this limitation via *CoSFan* has the potential to advance the state-of-the-art in CML.

Comparing computational efficiency in Table 1, the feed-forward method significantly outperformed MAML in both training and adaptation times across all continual strategies. While task-aware MAML had a linear slowdown in adaptation per task present, the feed-forward's parallelized forward pass was unaffected.

Table 1: CML adaptation efficiency comparison. All methods were adapted on 1200 batches. MAML-X refers to the X number of inner-steps.

| Model | Metric | T-Agnostic | T-Aware |
|---|---|---|---|
| MAML-1 | TTA-1 [s] | *0.0148(0.0009)* | *0.0150(0.0080)* |
| | TTA-12 [s] | *0.0149(0.0069)* | *0.1602(0.0058)* |
| | TTT [min] | *38.8(0.2)* | *116.3(11.0)* |
| MAML-5 | TTA-1 [s] | 0.0618(0.0077) | 0.0670(0.0096) |
| | TTA-12 [s] | 0.0636(0.0074) | 0.7475(0.0413) |
| | TTT [min] | 91.5(0.7) | 302.8(22.7) |
| FF | TTA-1 [s] | **0.0018(0.0048)** | **0.0017(0.0043)** |
| | TTA-12 [s] | **0.0018(0.0043)** | **0.0017(0.0005)** |
| | TTT [min] | **30.0(0.3)** | **85.2(5.5)** |

### 5.4 TASK-AWARE VS. TASK-RELATIONAL BUFFERS

*CoSFan* trained with the Task-Relational strategy showed a small dip in RP (averaging 20.7%) compared to the Task-Aware approach, remaining significantly better than all other baselines (full numerical results are in Appendix B.3). The performance drop may be due to some samples being misclassified into the wrong dynamics cluster, leading to incorrect context sets. This came with a benefit of automatically inferring relationships between all encountered tasks. Fig. 5A illustrates the reservoir's context embeddings and clusters over time, showing both aligned well with the underlying dynamics. Two Hamiltonian systems have their three parameter sets collapse into 1-2 clusters, suggesting the optimization found them insufficiently distinct to warrant separate clusters.

**Effect of Rare Tasks.** We expect task relation distillation to scale more effectively with numerous boundaries or recurring tasks. Fig. 5B shows results from an example where the first task appeared once, and three other tasks cycled through 40 boundaries, with the reservoir limited to 500 samples. The Task-Aware approach discarded most of the first task's data, retaining only 15 samples by the end, leading to deteriorating performance on this task. In contrast, the Task-Relational approach retained an even distribution of 125 samples per task, retaining stable rare-task performance.

## 6 CONCLUSIONS & DISCUSSION

We introduced *CoSFan*, a CML framework of *what-how & when*, which adapts latent neural dynamics to non-stationary distributions of dynamical systems - without task boundaries or identifiers. Additional ablation results show robustness to the size of $k$-shot context set (Appendix C.4), learning rate modulation (Appendix C.5), and increasingly restricted reservoir sizes (Appendix C.3). We show a resilience to gradual task shifts (Appendix C.8) . We also justify the task boundary threshold $\nu$ (Appendix C.7) and use of cluster embedding regularization (Appendix C.6). **Limitations.** Testing on real-world datasets and on broader CML tasks (e.g., meta-reinforcement learning) is needed to support our findings. The assumption of *local stationarity* may limit some applications.

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

## A   RELATED WORKS

In addition to learning-to-learn latent dynamics and CML, the proposed work is also relevant to the areas of research in continual learning and meta-continual learning.

**Continual-learning.**   Relevant to these limitations is the research area of *continual learning*, in which the fundamental goal is to handle data non-stationarity as the ability to simultaneously preserve old performance while accelerating the aggregation of new knowledge. Under this framework, incoming data is seen as a sequence of distinct tasks expressing unique properties that differentiate themselves (Van de Ven & Tolias, 2019). A multitude of approaches have been proposed under continual learning to alleviate catastrophic forgetting, with the most representative approaches categorized as: *i) rehearsal-based* methods that approximate the stationary training distribution via storing or generating samples over time (Robins, 1995), *ii) structural* methods that preserve sub-networks of neural networks over time to maintain per-task performance (Rusu et al., 2016), and *iii) regularization-based* methods that leverage the sequentially learned Bayesian posteriors as priors that constrain learning on new tasks (He et al., 2019).

However, an underlying assumption of these continual methods is that, regardless of what tasks have been or are being learned, the network is able to learn a set of parameters that can perform well across all tasks. In heterogeneous time-series forecasting, this is often an invalid assumption as the input-output mapping can conflict as the same observation can provide different forecasts depending on the (potentially unknown) system context. As such, naïve continual learning is not directly applicable to this setting as it lacks both task-inference and context-aware adaptation mechanisms.

**Meta-continual learning.**   The recent field of *meta-continual learning* (MCL), though similar in name and approaches considered, is distinct from CML in its goal and optimization. CML uses continual learning techniques to enable the training of meta-learning models over a distribution of tasks that appears sequentially over time. Its goal is to preserve the performance of the meta-model on the previous tasks, even when little-to-no data for those tasks are available, while also aggregating knowledge from the current task. Conversely, MCL aims to frame the continual learning problem as a meta-learning problem, in which each meta-task is some variation on the order that the continual learning tasks are presented in. Its goal is to meta-learn the continual learning algorithm such that on a new sequence of tasks, it can adapt quickly. Unlike CML, which has context-dependent targets, MCL assumes a fixed target distribution of some x-y mapping. Due to differences in the goal of the meta-models and the data settings considered, we do not consider MCL relevant to our evaluation and omit comparisons against their methods.

**Summary of related work.**   The terminology and goals of meta-continual learning and continual meta-learning are nuanced and often challenging to distinguish, as both adapt the same fundamental methods to different ends. We defer to the breakdown in (Caccia et al., 2020) for a formalized review of MCL and CML, and how they relate to meta-learning and continual-learning, respectively.

**Alternative algorithmic priors.**   Beyond gradient-based and feed-forward hyper-networks, sequence learners such as Transformers (Vaswani, 2017) are powerful architectures for extracting knowledge in high-dimensional settings. In standard meta-learning, Transformers have been used as algorithmic priors (Chen & Wang, 2022) by combining an initial token set of parameters with tokenized context data to adapt the weights, a conceptually similar approach to gradient-based meta-learners' initial parameter set. A recent CML work (Vladymyrov et al.) further explores using Transformer architectures as hyper-networks to generate target network weights based on the context set. In their framework, the weights generated for a previous task are reused as parameter tokens to update the current task's weights alongside active task samples. Notably, this method omits the use of a replay buffer, relying instead on weights updated iteratively through active samples. While this approach primarily targets image classification tasks, adapting it to latent dynamics settings offers a promising direction for future exploration.

In this work, we focus on applying feed-forward algorithmic priors, specifically in the form of a hyper-network, and comparing their performance to the standard CML approach of gradient-based meta-learners. However, exploring other algorithmic priors represents an intriguing avenue for future research.

# B FULL QUANTITATIVE RESULTS

## B.1 COMPLETE SECTION 5.2 RESULTS

Here we include the full numerical experimental results for the comparison to existing latent dynamics baselines on *gravity-6* and *mixed-physics* in Table 2 and Table 3, respectively. The numerical results of these methods trained under the stationary distribution, where all tasks are available simultaneously and have known task identifiers, are shown in Table 4.

As well, we include a similar visualization as in Fig. 2 but for MSE on **mixed-physics** in Fig. 6 and for DST and MSE on **gravity-6** in Fig. 7 and Fig. 8, respectively.

Table 2: Performance comparison on *gravity-6* over existing latent dynamics models and their continual extensions. All methods were run over 5 seeds.

| Model | Metric | Naive | | Exact Replay | | Boundary Reservoir | |
|---|---|---|---|---|---|---|---|
| | | *DST* | *MSE* | *DST* | *MSE* | *DST* | *MSE* |
| DKF | LP | 9.13(1.07) | 0.0234(0.0003) | 6.39(0.07) | 0.0219(0.0001) | 7.98(1.06) | 0.0228(0.0004) |
| | RP | 8.80(0.77) | 0.0234(0.0003) | 6.79(0.11) | 0.0222(0.0001) | 7.13(0.47) | 0.0226(0.0003) |
| | BTI | 0.33(0.30) | 0.0000(0.0000) | -0.39(0.15) | -0.0003(0.0001) | 0.85(0.97) | 0.0002(0.0003) |
| VRNN | LP | 9.17(0.27) | 0.0221(0.0000) | 4.69(0.18) | 0.0150(0.0001) | 7.49(0.71) | 0.0190(0.0011) |
| | RP | 10.07(0.51) | 0.0215(0.0000) | 8.99(0.17) | 0.0214(0.0002) | 9.08(0.78) | 0.0225(0.0018) |
| | BTI | -0.91(0.27) | 0.0006(0.0000) | -4.30(0.19) | -0.0065(0.0003) | -1.59(0.57) | -0.0035(0.0015) |
| RGNRes | LP | 9.23(0.52) | 0.0195(0.0002) | 2.73(0.16) | 0.0129(0.0002) | 3.82(0.53) | 0.0146(0.0009) |
| | RP | 12.63(1.30) | 0.0203(0.0001) | 7.83(0.17) | 0.0199(0.0005) | 7.48(0.20) | 0.0214(0.0008) |
| | BTI | -3.40(1.45) | -0.0009(0.0001) | -5.10(0.15) | -0.0071(0.0005) | -3.66(0.34) | -0.0068(0.0010) |
| MetaRGNRes | LP | 9.92(0.32) | 0.0199(0.0002) | 1.62(0.08) | 0.0111(0.0002) | 1.67(0.09) | 0.0114(0.0001) |
| | RP | 13.04(1.22) | 0.0207(0.0001) | 1.57(0.03) | 0.0108(0.0001) | 1.72(0.08) | 0.0120(0.0002) |
| | BTI | -3.12(1.31) | -0.0008(0.0001) | 0.05(0.06) | 0.0002(0.0002) | -0.05(0.07) | -0.0006(0.0003) |

Table 3: Performance comparison on *mixed-physics* over existing latent dynamics models and their continual extensions. All methods were run over 5 seeds.

| Model | Metric | Naive | | Exact Replay | | Boundary Reservoir | |
|---|---|---|---|---|---|---|---|
| | | *DST* | *MSE* | *DST* | *MSE* | *DST* | *MSE* |
| DKF | LP | 2.73(0.39) | 0.0228(0.0013) | 2.52(0.24) | 0.0220(0.0006) | 2.93(0.41) | 0.0231(0.0006) |
| | RP | 4.49(0.33) | 0.0445(0.0110) | 2.78(0.31) | 0.0223(0.0008) | 2.84(0.31) | 0.0226(0.0007) |
| | BTI | -1.76(0.66) | -0.0217(0.0115) | -0.15(0.03) | -0.0003(0.0003) | 0.09(0.12) | 0.0005(0.0005) |
| VRNN | LP | 2.48(0.23) | 0.0164(0.0010) | 2.71(0.38) | 0.0169(0.0007) | 2.97(0.51) | 0.0182(0.0009) |
| | RP | 4.87(1.01) | 0.0434(0.0034) | 3.70(0.45) | 0.0214(0.0006) | 3.78(0.52) | 0.0234(0.0005) |
| | BTI | -2.50(0.84) | -0.0270(0.0034) | -0.99(0.19) | -0.0045(0.0008) | -0.81(0.21) | -0.0052(0.0011) |
| RGNRes | LP | 2.43(0.82) | 0.0162(0.0022) | 1.56(0.00) | 0.0143(0.0000) | 1.92(0.33) | 0.0162(0.0010) |
| | RP | 4.75(0.32) | 0.0578(0.0141) | 2.94(0.00) | 0.0188(0.0000) | 2.70(0.30) | 0.0208(0.0008) |
| | BTI | -1.87(0.91) | -0.0415(0.0120) | -1.38(0.00) | -0.0045(0.0000) | -0.78(0.07) | -0.0046(0.0005) |
| MetaRGNRes | LP | 1.32(0.00) | 0.0128(0.0000) | 1.54(0.07) | 0.0139(0.0007) | 1.56(0.14) | 0.0141(0.0007) |
| | RP | 4.52(0.00) | 0.0736(0.0000) | 1.59(0.08) | 0.0140(0.0009) | 1.61(0.13) | 0.0154(0.0008) |
| | BTI | -3.19(0.00) | -0.0608(0.0000) | -0.05(0.09) | -0.0001(0.0003) | -0.05(0.02) | -0.0013(0.0002) |

Table 4: Performance comparison on **gravity-6** and **mixed-physics** over existing latent dynamics models trained on the true stationary distribution. All methods were run over 5 seeds.

| Model | gravity-6 | | mixed-physics | |
|---|---|---|---|---|
| | *DST* | *MSE* | *DST* | *MSE* |
| DKF | 6.41(0.00) | 0.0220(0.0000) | 4.14(0.03) | 0.0220(0.0001) |
| VRNN | 8.86(0.14) | 0.0192(0.0000) | 4.37(0.05) | 0.0201(0.0001) |
| RGNRes | 7.37(0.12) | 0.0185(0.0001) | 2.37(0.02) | 0.0203(0.0001) |
| MetaRGNRes | 3.51(0.45) | 0.0197(0.0011) | 2.34(0.23) | 0.0194(0.0011) |

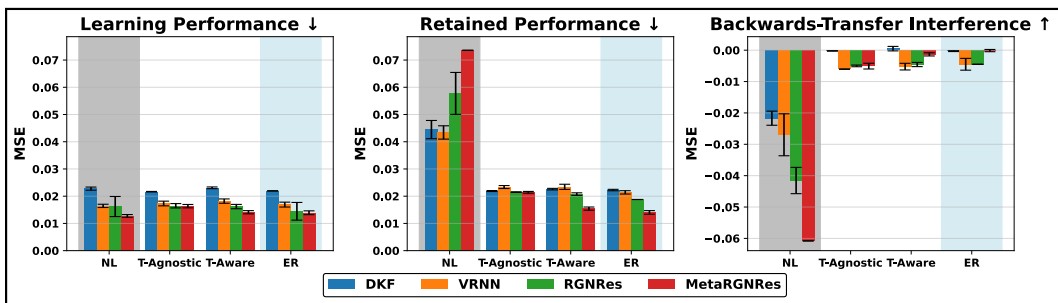

Figure 6: MSE performance comparison on *mixed-physics* over existing latent dynamics models and their continual extensions. All methods were run over 5 seeds.

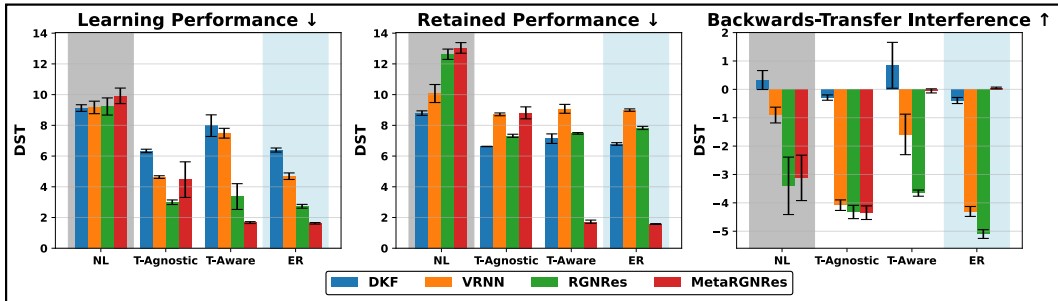

Figure 7: DST performance comparison on *gravity-6* over existing latent dynamics models and their continual extensions. All methods were run over 5 seeds.

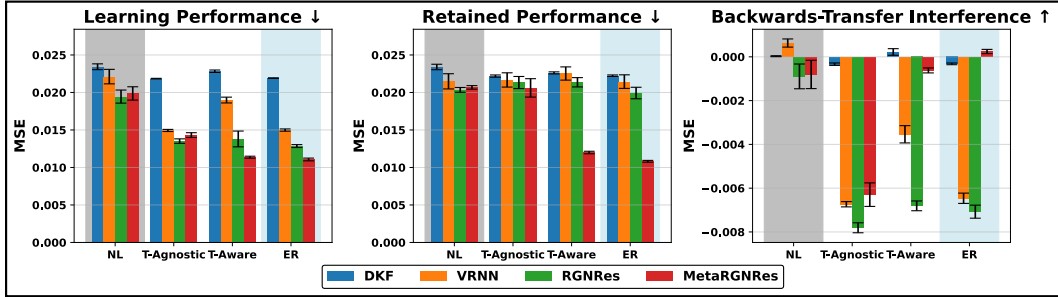

Figure 8: MSE performance comparison on *gravity-6* over existing latent dynamics models and their continual extensions. All methods were run over 5 seeds.

## B.2 COMPLETE SECTION 5.3 RESULTS

Here we include the full numerical experimental results for the comparison to existing CML formulations on *gravity-6* and *mixed-physics* in Table 6 and Table 5, respectively. Table 7 shows the complete efficiency comparison, including BGD.

As well, we include a similar visualization as in Fig. 4 but for DST and MSE on **mixed-physics** in Fig. 9 and Fig. 10, respectively, and for DST on **gravity-6** in Fig. 11.

Table 5: Performance comparison on **mixed-physics** over existing CML formulations and the feed-forward approach. All methods were run over 5 seeds.

| Model | Metric | BGD | | Task-Agnostic | | Task-Aware | |
|---|---|---|---|---|---|---|---|
| | | *DST* | *MSE* | *DST* | *MSE* | *DST* | *MSE* |
| Feed-Forward | LP | 10.92(0.00) | 0.0563(0.0000) | 1.76(0.21) | 0.0162(0.0014) | 1.56(0.14) | 0.0141(0.0007) |
| | RP | 8.50(0.00) | 0.0563(0.0000) | 2.70(0.20) | 0.0219(0.0012) | 1.61(0.13) | 0.0154(0.0008) |
| | BTI | 2.42(0.00) | -0.0000(0.0000) | -0.94(0.01) | -0.0057(0.0003) | -0.05(0.02) | -0.0013(0.0002) |
| MAML [1-Step] | LP | 6.66(1.18) | 0.0505(0.0039) | 1.83(0.20) | 0.0164(0.0009) | 2.68(0.13) | 0.0182(0.0008) |
| | RP | 5.90(0.59) | 0.0516(0.0037) | 2.67(0.34) | 0.0208(0.0011) | 2.62(0.14) | 0.0185(0.0010) |
| | BTI | 0.77(0.99) | -0.0011(0.0010) | -0.84(0.17) | -0.0044(0.0007) | 0.06(0.07) | -0.0003(0.0003) |
| MAML [5-Step] | LP | 9.60(4.38) | 0.0503(0.0045) | 2.18(0.35) | 0.0176(0.0009) | 3.07(0.58) | 0.0192(0.0008) |
| | RP | 10.70(6.25) | 0.0515(0.0036) | 2.81(0.40) | 0.0207(0.0014) | 2.92(0.43) | 0.0186(0.0007) |
| | BTI | -1.10(1.96) | -0.0012(0.0020) | -0.63(0.11) | -0.0031(0.0008) | 0.15(0.22) | 0.0006(0.0004) |

Table 6: Performance comparison on **gravity-6** over existing CML formulations and the feed-forward approach. All methods were run over 5 seeds.

| Model | Metric | BGD | | Task-Agnostic | | Task-Aware | |
|---|---|---|---|---|---|---|---|
| | | *DST* | *MSE* | *DST* | *MSE* | *DST* | *MSE* |
| Feed-Forward | LP | 8.22(0.54) | 0.0235(0.0004) | 4.47(3.39) | 0.0143(0.0019) | 1.67(0.09) | 0.0114(0.0001) |
| | RP | 11.11(0.33) | 0.0243(0.0001) | 8.81(2.65) | 0.0206(0.0007) | 1.72(0.08) | 0.0120(0.0002) |
| | BTI | -2.90(0.72) | -0.0009(0.0004) | -4.35(0.79) | -0.0063(0.0022) | -0.05(0.07) | -0.0006(0.0003) |
| MAML [1-Step] | LP | 8.57(1.27) | 0.0236(0.0007) | 2.12(0.05) | 0.0129(0.0002) | 5.34(0.24) | 0.0165(0.0005) |
| | RP | 10.68(0.53) | 0.0244(0.0001) | 7.24(0.18) | 0.0199(0.0003) | 5.94(0.44) | 0.0174(0.0004) |
| | BTI | -2.11(1.37) | -0.0008(0.0007) | -5.12(0.16) | -0.0070(0.0003) | -0.60(0.41) | -0.0009(0.0007) |
| MAML [5-Step] | LP | 7.70(0.76) | 0.0232(0.0006) | 2.56(0.48) | 0.0137(0.0006) | 4.67(1.89) | 0.0156(0.0018) |
| | RP | 10.19(0.87) | 0.0242(0.0005) | 6.81(0.14) | 0.0198(0.0007) | 5.31(2.43) | 0.0156(0.0017) |
| | BTI | -2.49(0.79) | -0.0010(0.0006) | -4.25(0.43) | -0.0061(0.0002) | -0.64(0.76) | -0.0000(0.0005) |

Table 7: Complete adaptation efficiency comparison on the choice of meta-model and continual-strategy. All methods were adapted over 100 batches of data across the 12 **mixed-physics'** dynamics, timing each batch independently. **Bold** and *italic* indicate 1st- and 2nd-best metrics, respectively.

| Model | Metric | BGD | Task-Agnostic Reservoir | Task-Aware Reservoir |
|---|---|---|---|---|
| MAML [1-Step] | TTA [1] (seconds) | *0.0155(0.0011)* | *0.0148(0.0009)* | *0.0150(0.0080)* |
| | TTA [12] (seconds) | *0.0152(0.0013)* | *0.0149(0.0069)* | *0.1602(0.0058)* |
| | TTT (minutes) | *126.3(4.7)* | *38.8(0.2)* | *116.3(11.0)* |
| MAML [5-Step] | TTA [1] (seconds) | 0.0637(0.0025) | 0.0618(0.0077) | 0.0670(0.0096) |
| | TTA [12] (seconds) | 0.0623(0.0015) | 0.0636(0.0074) | 0.7475(0.0413) |
| | TTT (minutes) | 315.8(6.5) | 91.5(0.7) | 302.8(22.7) |
| Feed-Forward | TTA [1] (seconds) | **0.0017(0.0005)** | **0.0018(0.0048)** | **0.0017(0.0043)** |
| | TTA [12] (seconds) | **0.0018(0.0006)** | **0.0018(0.0043)** | **0.0017(0.0005)** |
| | TTT (minutes) | **122.1(3.4)** | **30.0(0.3)** | **85.2(5.5)** |

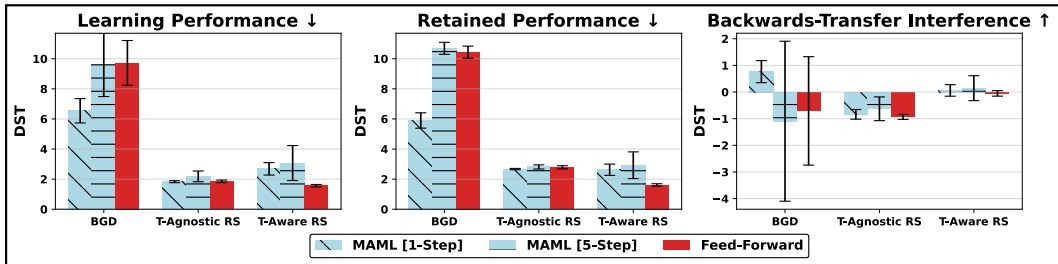

Figure 9: DST performance comparison on **_mixed-physics_** over existing CML formulations and the feed-forward approach. All methods were run over 5 seeds.

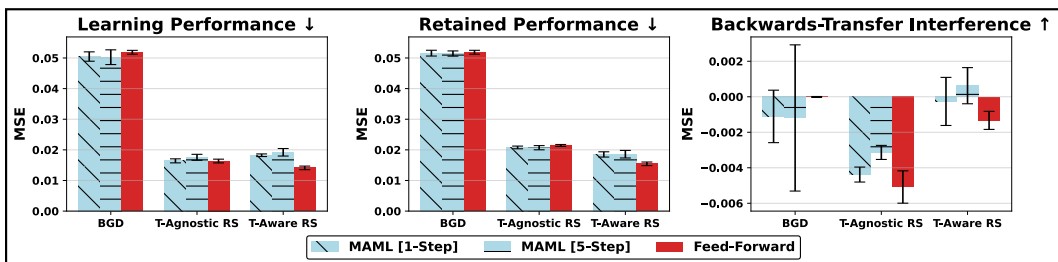

Figure 10: MSE performance comparison on **_mixed-physics_** over existing CML formulations and the feed-forward approach. All methods were run over 5 seeds.

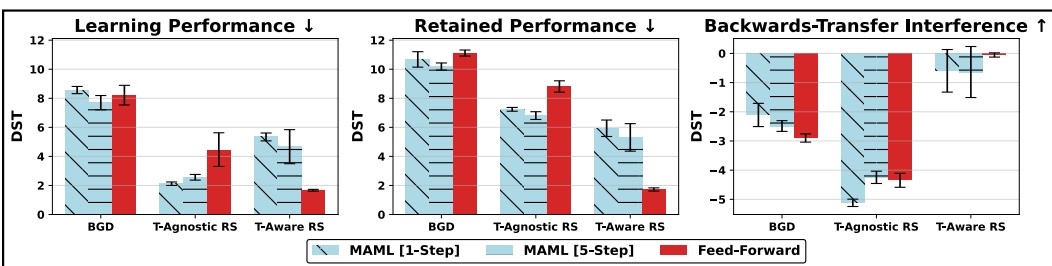

Figure 11: DST performance comparison on **_gravity-6_** over existing CML formulations and the feed-forward approach. All methods were run over 5 seeds.

### B.3 COMPLETE SECTION 5.4 RESULTS

Here we include the full numerical experimental results for the comparison between the Task-Aware Reservoir Sampling methods and the Task-Relational Experience Replay on *gravity-6* and *mixed-physics* in Table 8. As well, Fig. 12 and Fig. 13 show the per-task performance of both continual strategies when continually learning over the sequentially presented tasks, on **mixed-physics** and **gravity-6** respectively.

**Best Practice.** We suggest using the Task-Relational approach under limited memory scenarios, or when there is prior knowledge about frequent switching among recurring tasks or underrepresentation of rare tasks. In settings where it is assumed only novel tasks will show up over time, the Task-Aware method is sufficient to identify the tasks and comes with better training efficiency.

Table 8: Comparison between Task-Aware Reservoir Sampling and Task-Relational Experience Replay. 5 seeds were tested. **Bold** and *italic* indicate 1st- and 2nd-best metrics, respectively.

| Model | Metric | Task-Aware Reservoir Sampling | | Task-Relational Experience Replay | |
| | | *DST* | *MSE* | *DST* | *MSE* |
|---|---|---|---|---|---|
| **mixed-physics** | LP | **1.56(0.14)** | **0.0141(0.0007)** | *1.61(0.12)* | 0.0143(0.0008) |
| | RP | **1.61(0.13)** | **0.0154(0.0008)** | *1.91(0.22)* | 0.0172(0.0012) |
| | BTI | **-0.05(0.02)** | **-0.0013(0.0002)** | *-0.29(0.14)* | -0.0029(0.0007) |
| **gravity-6** | LP | *1.67(0.09)* | **0.0114(0.0001)** | **1.67(0.07)** | *0.0115(0.0002)* |
| | RP | **1.72(0.08)** | **0.0120(0.0002)** | 2.11(0.26) | *0.0128(0.0006)* |
| | BTI | **-0.05(0.07)** | **-0.0006(0.0003)** | *-0.44(0.31)* | *-0.0013(0.0007)* |

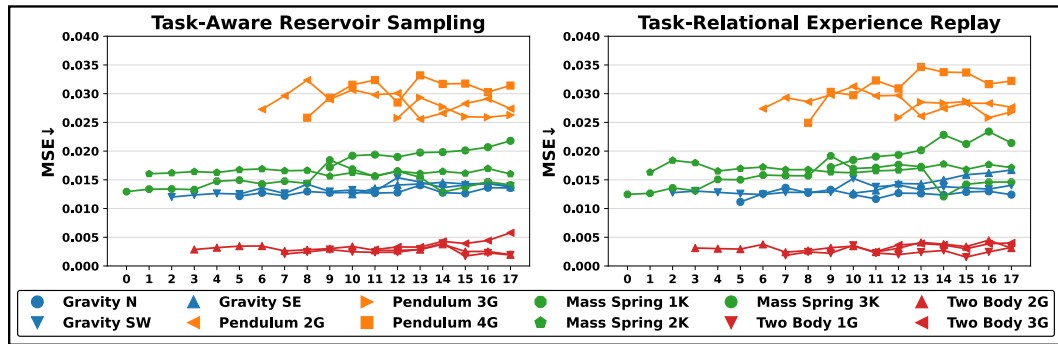

Figure 12: Performance comparison per-task over the continual sequence between Task-Aware Reservoir Sampling and Task-Relational Experience Replay evaluated on *mixed-physics*.

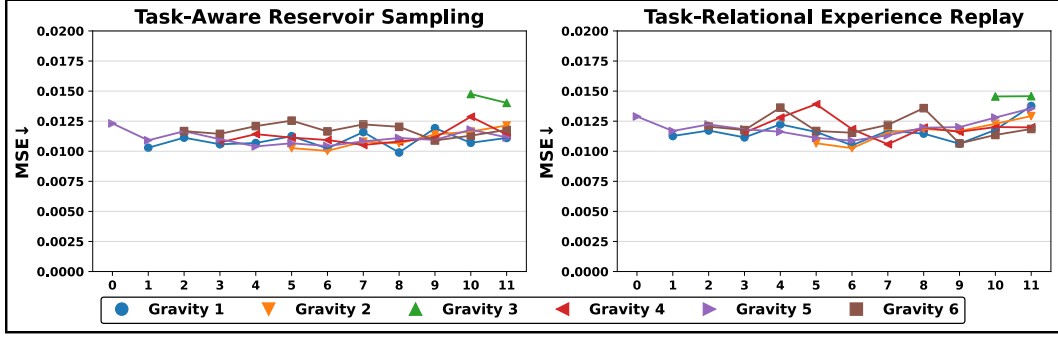

Figure 13: Performance comparison per-task over the continual sequence between Task-Aware Reservoir Sampling and Task-Relational Experience Replay evaluated on *gravity-6*.

## C    ADDITIONAL ABLATION RESULTS

### C.1    ADDITIVE VS. MULTIPLICATIVE INTERACTIONS

The original meta-sLVM formulation proposed by (Jiang et al., 2023) considers an additive conditioning mechanism, where the derived context variable $\mathbf{c}(\mathcal{T}_j)$ conditions the latent dynamics function by concatenating it with the latent state, represented as $\mathbf{z}_0^* = [\mathbf{z}_0, \mathbf{c}(\mathcal{T}_j)]$, before propagating the dynamics. In this work, we choose the hyper-network conditioning mechanism instead, as prior evaluations of conditioning interactions suggest that multiplicative interactions, such as those enabled by hyper-networks, are a super-set of additive interactions and provide stronger inductive biases for a broader range of function families (Jayakumar et al., 2020).

Table 9 compares the performance of the additive mechanism with the proposed hyper-network conditioning method, both evaluated using Task-Aware Reservoir Sampling on ***mixed-physics***. Fig. 14 highlights the progression of the tasks' metrics over the course of training between both methods. The results indicate comparable performance between the two methods, suggesting that the considered baselines do not require the additional flexibility of multiplicative interactions, and that this flexibility does not hinder convergence.

Table 9: Performance comparison between the additive conditioning mechanism from (Jiang et al., 2023) and the proposed hyper-network conditioning mechanism evaluated on *mixed-physics*

| Method | *Learned* DST↓ | *Learned* MSE↓ | *Retained* DST↓ | *Retained* MSE↓ | *Interference* DST↓ | *Interference* MSE↓ |
|---|---|---|---|---|---|---|
| Hypernet | 1.56(0.14) | 0.0141(0.0007) | 1.61(0.13) | 0.0154(0.0008) | -0.05(0.02) | -0.0013(0.0002) |
| Additive | 1.61(0.08) | 0.0145(0.0006) | 1.62(0.10) | 0.0155(0.0009) | -0.01(0.08) | -0.0010(0.0004) |

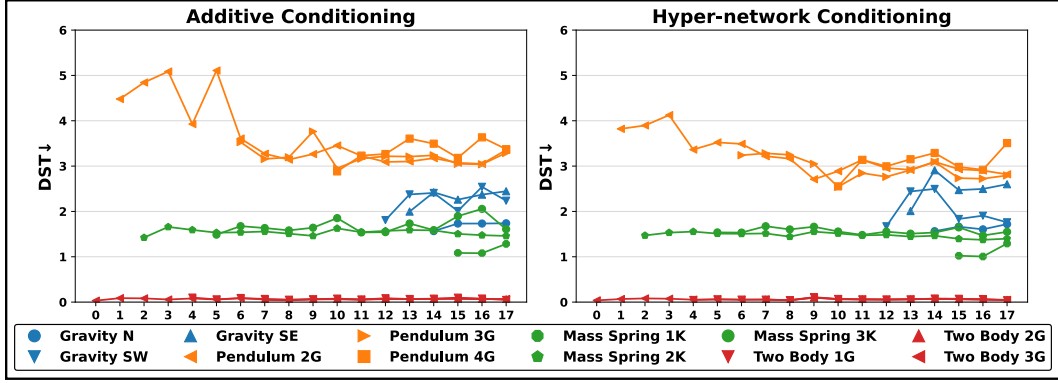

Figure 14: Performance comparison per-task over the continual sequence between the additive conditioning mechanism from (Jiang et al., 2023) and the proposed hyper-network conditioning mechanism evaluated on *mixed-physics*.

## C.2 COMPUTATIONAL REQUIREMENTS OF TASK-RELATIONAL EXPERIENCE REPLAY

In this section, we evaluate the computational and memory requirements of Task-Relational Experience Replay during training. Notably, this replay strategy does not impact adaptation efficiency at test time and introduces additional computational demands only at detected task boundaries. For fitting the Bayesian Gaussian Mixture Model, we utilize the scikit-learn library due to its robust implementation and feature set. This process is performed on the CPU, requiring data transfer between the GPU and CPU for the meta-embeddings, which may introduce a minor slowdown in the presented results here. While a GPU-based implementation would eliminate this transfer and provide computational speed-ups, we note that the current CPU-based fitting remains efficient, with execution times within seconds for the datasets evaluated.

To assess the scalability of the Bayesian Gaussian Mixture Model with respect to task numbers and reservoir size, we varied the reservoir size from 500 to 4500 samples in increments of 1000. For each configuration, over a range of 12 unique dynamics, we measured the average mixture model fitting time (in seconds), the average memory usage during fitting (in MB), and the average memory usage after fitting (in MB). Fig. 15 presents these metrics when applied over 5 seeds. The results indicate favorable scalability in both memory and speed requirements as the reservoir size and number of unique dynamics increase.

We measured the Time-to-Train metric for the Task-Relational Experience Replay mechanism and compared it to the Task-Aware Reservoir Sampler on the ***mixed-physics*** dataset, using a reservoir size of 4500 samples and averaging results over 5 seeds. The Task-Aware mechanism required $83.04 \pm 0.67$ minutes to train, while the Task-Relational mechanism required $85.20 \pm 2.25$ minutes, indicating a relatively minor increase of $2.5\%$ in training time for the additional benefits provided by task-relational modeling.

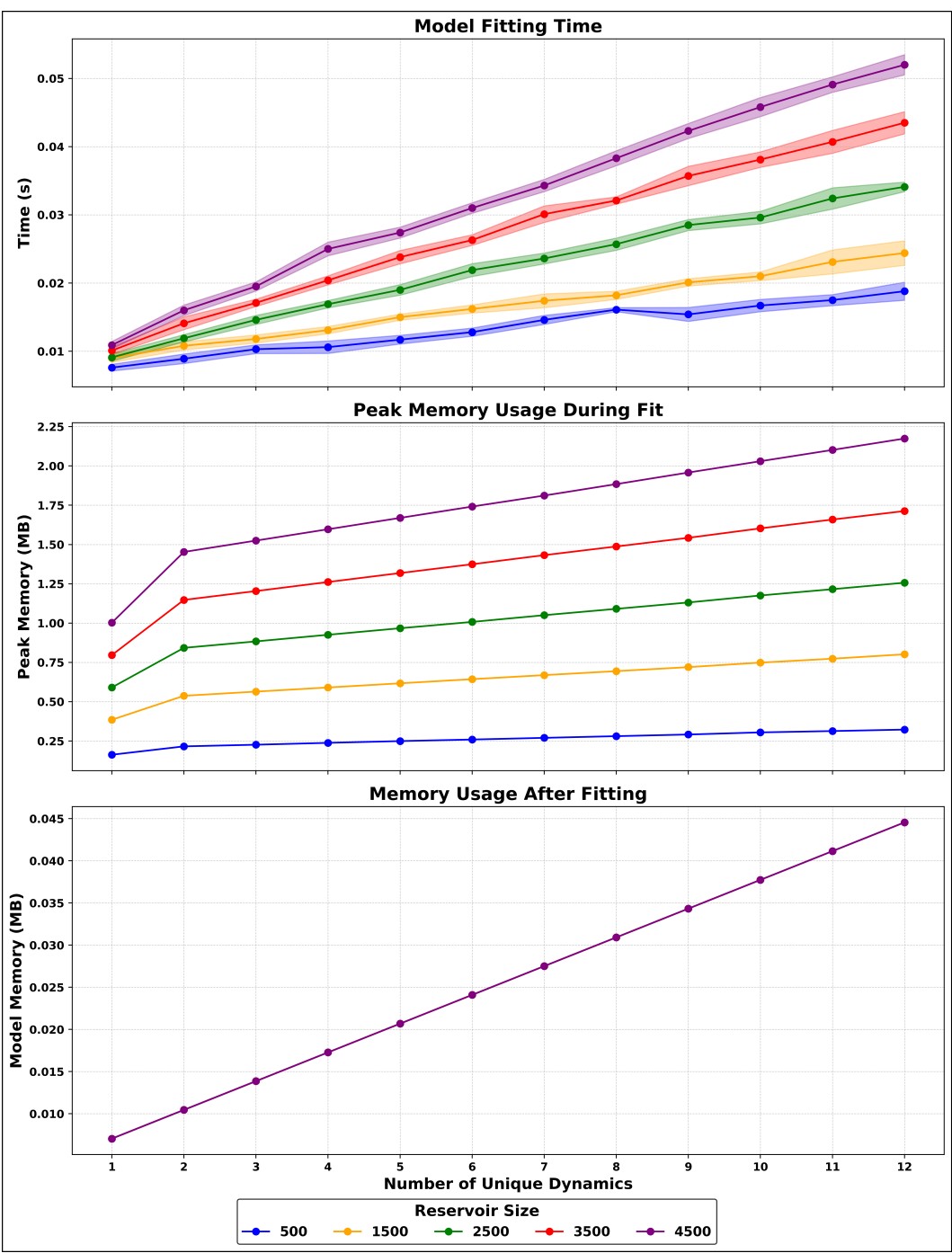

Figure 15: Memory and speed performance ablation on the Task-Relational Experience Replay mechanism over increasing reservoir sizes and number of unique dynamics.

## C.3 EFFECT OF RESERVOIR SIZE

We investigated the task-aware feed-forward method across increasing reservoir sizes on **mixed-physics** to understand the extent to which it can preserve performance as less and less data becomes available for prior tasks. The results are shown in Fig. 16 for MSE and Fig. 17 for DST . It can be seen that while Learning Performance remains similar across the reservoir sizes, the Retained Performance and catastrophic forgetting exhibited lessens up to a certain point before beginning to stabilize at the higher numbers.

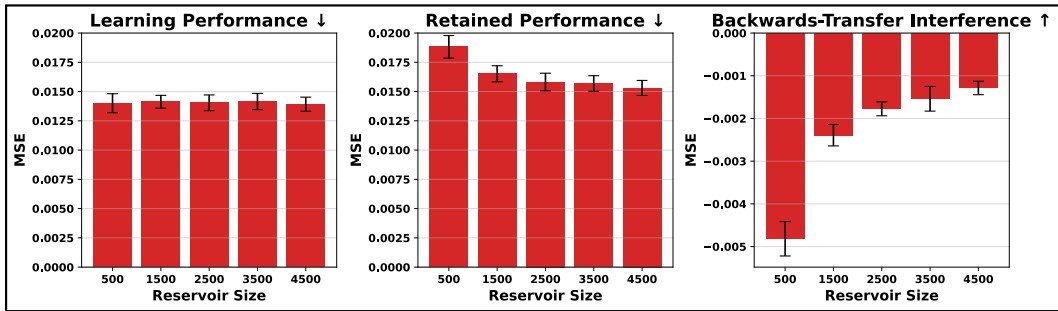

Figure 16: MSE performance of the task-aware feed-forward meta-model on varying reservoir sizes for *mixed-physics*.

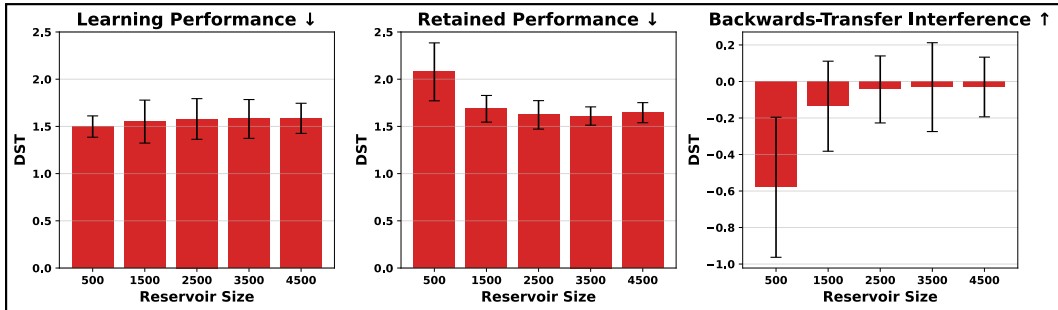

Figure 17: DST performance of the task-aware feed-forward meta-model on varying reservoir sizes for *mixed-physics*.

## C.4 EFFECT OF CONTEXT-SET $k$

We performed an ablation for the feed-forward model with task-aware reservoir sampling over increasing $k$-shot context set sizes on **mixed-physics**, shown in Fig. 18 for MSE and Fig. 19 for DST . A slight trend of improvement can be seen in the Retained Performance as the number of samples in the context set increases, though notably it is the variance exhibited in the Learning Performance and Backwards-Transfer Inference of DST that decreases at higher k-shots.

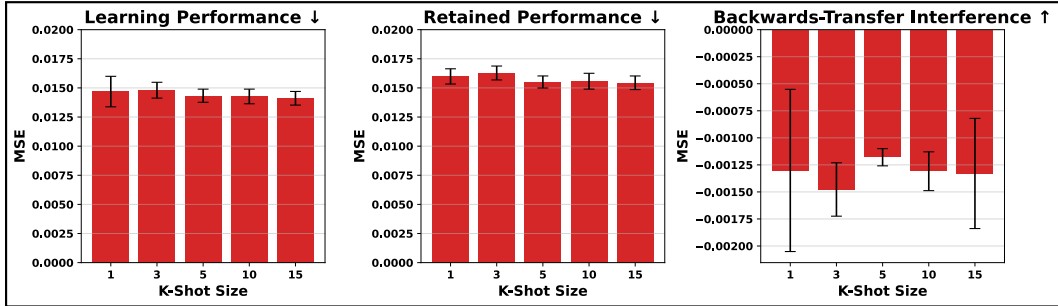

Figure 18: MSE performance of the task-aware feed-forward meta-model on varying $k$-shot sizes for *mixed-physics*.

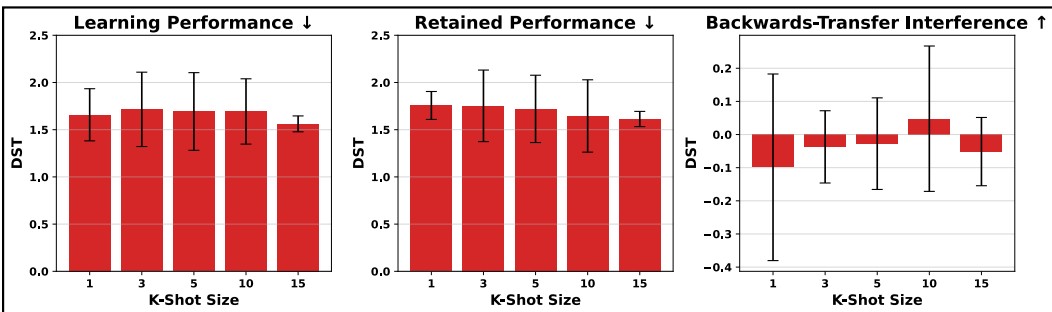

Figure 19: DST performance of the task-aware feed-forward meta-model on varying $k$-shot sizes for *mixed-physics*.

### C.5 Effect of Update Modulation $um$

Learning rate schedulers are often essential for achieving good convergence in deep learning (Botev et al., 2021). However, this presents a unique challenge in settings where task boundaries are unknown. While *local stationarity* could guide scheduling, available strategies (i.e., fully reducing the learning rate by the end of a window or maintaining a higher learning rate) pose potential optimization risks. Inspired by the update modulation schema described in (Caccia et al., 2020), we tested using a dynamic modulation coefficient that scales the learning rate based on the loss incurred at each timestep. This coefficient is calculated using the equation $um = 1 - e^{-\upsilon x}$, where the hyper-parameter $\upsilon$ adjusts the modulation's likelihood scale from 0 to 1.

Results of an ablation over varying values for $\upsilon$ are presented in Fig. 20 and Fig. 21 for MSE and DST, respectively. Interestingly, we found that neither the presence nor value of the update modulator had much of an effect on overall convergence. We attribute this to the datasets having a generally stable likelihood range from batch to batch as well as the choice of an adaptive learning rate may provide enough of a stabilization that an additional component on top shows little effect.

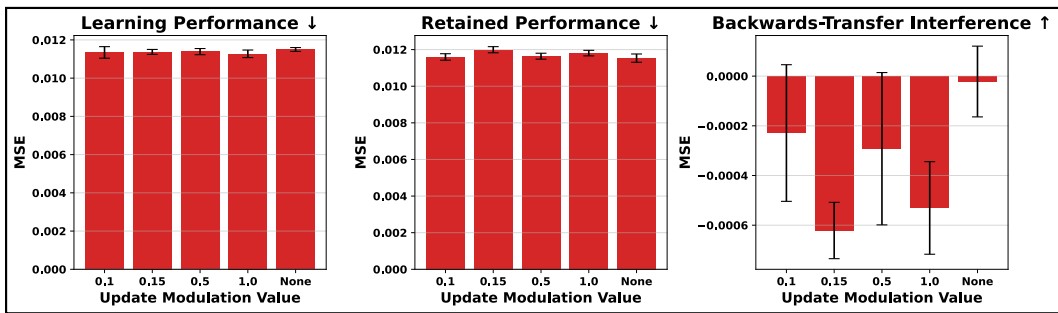

Figure 20: MSE performance of the task-aware feed-forward meta-model on varying update modulation $um$ values for *gravity-6*.

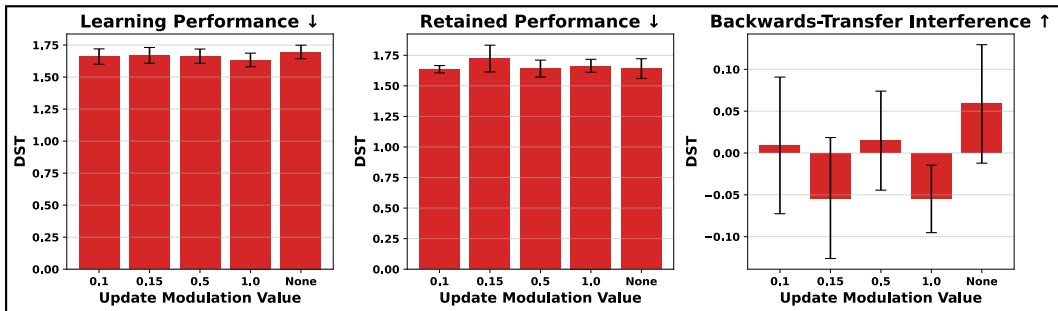

Figure 21: DST performance of the task-aware feed-forward meta-model on varying update modulation $um$ values for *gravity-6*.

## C.6 EFFECT OF CLUSTER REGULARIZATION

We performed an ablation study on the cluster regularization term for the feed-forward model with task-relational experience replay, evaluating its effect across a range of $\beta$ values in the loss function. Fig. 22 and Fig. 23 presents the numerical results for MSE and DST, respectively, on the **mixed-physics** task. The inclusion of the cluster regularization term stabilized the BTI of the meta-model across different seeds though left LP and RP mostly unaffected as the weight increased.

Fig. 24 visualizes the final reservoir context-embedding space for models with no regularization and with a regularization strength of 1e-2. The inclusion of the regularization term improved the separation of the Two-Body equations into more distinct clusters and, notably, disentangled the Gravity dynamics from the Pendulum system.

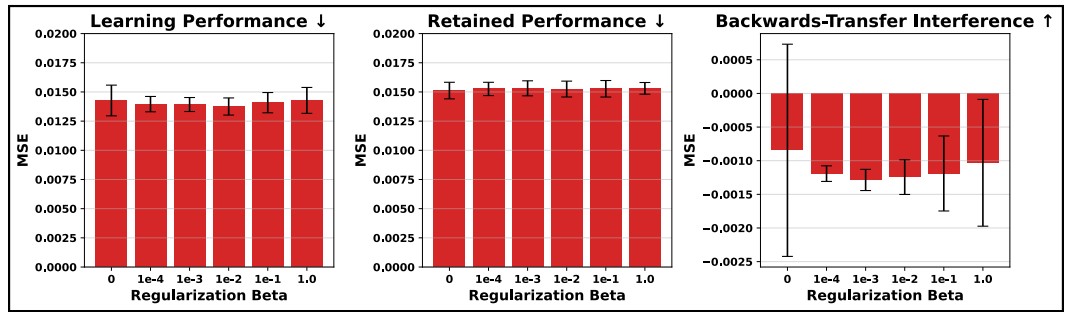

Figure 22: MSE performance of the task-relational feed-forward meta-model methods on varying cluster regularization $\beta$ values for *mixed-physics*.

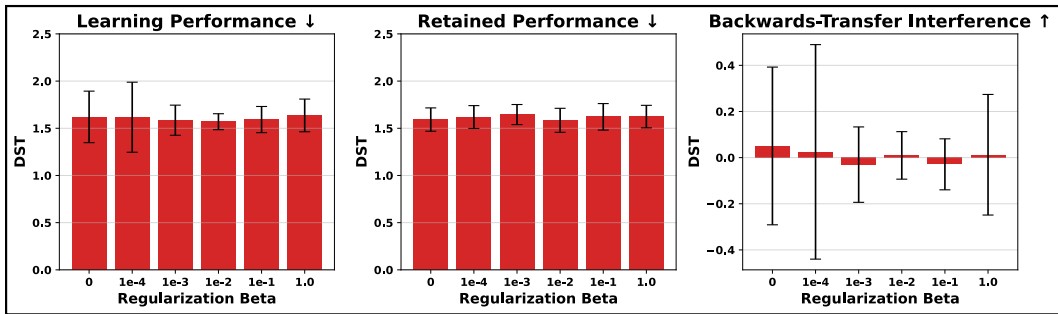

Figure 23: DST performance of the task-relational feed-forward meta-model methods on varying cluster regularization $\beta$ values for *mixed-physics*.

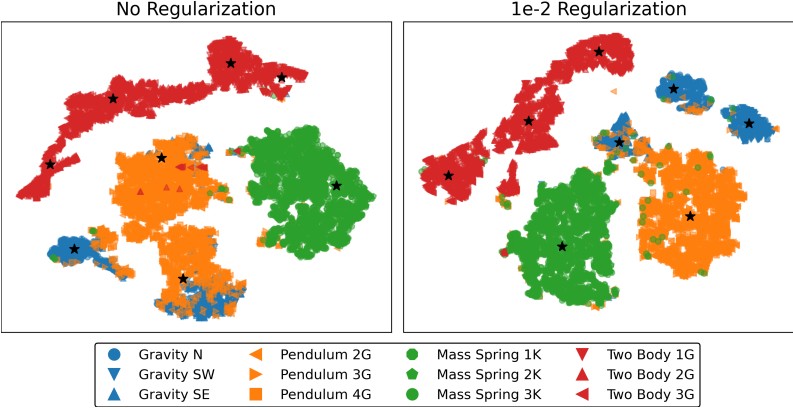

Figure 24: t-SNE visualization of the reservoir's resulting context-embedding space for a model with no cluster regularization (left) and with a $\beta$ of 1e-2 for cluster regularization (right).

### C.7 TASK BOUNDARY THRESHOLD $\nu$

We investigated the task boundary detection mechanism and justify the use of the threshold parameter, $\nu$. Fig. 25 and Fig. 26 show the progression of the task-aware feed-forward model's training likelihood across the tasks for both **gravity-6** and **mixed-physics** across the 5 seeds, respectively. Markers show the flagged performance dip on the task boundary where context and query sets stem from different origins. For both datasets, it is clear that a significant dip occurs at the task switch, making it feasible to use a task boundary detection method. We note that the choice of $\nu$ is dependent on the data distribution and expected likelihood ranges.

Rather than a static threshold value, it may be possible to instead maintain an online estimate of the training likelihood mean $\mu_{\mathcal{L}_j}$ and standard deviation $\sigma_{\mathcal{L}_j}$, flagging a new task whenever a likelihood comes in that is out-of-distribution via:

$$||\mathcal{L}_{j,n} - \mathcal{L}_{j,n-1}|| > \mu_{\mathcal{L}_j} + (2.5 * \sigma_{\mathcal{L}_j}) \tag{8}$$

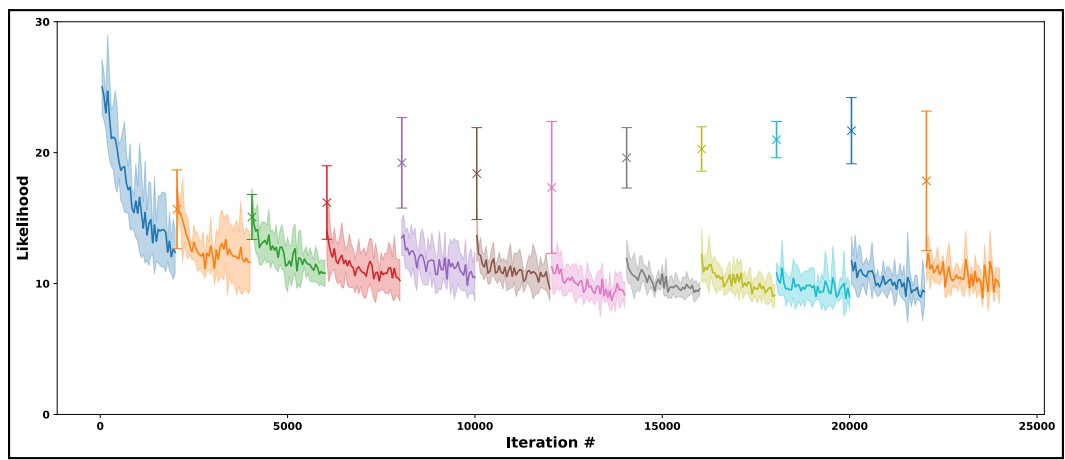

Figure 25: Progression of the feed-forward's training likelihood over tasks on **gravity-6**, highlighting via markers the performance dip exhibited at task boundaries across the seeds. Shaded areas represent one standard deviation.

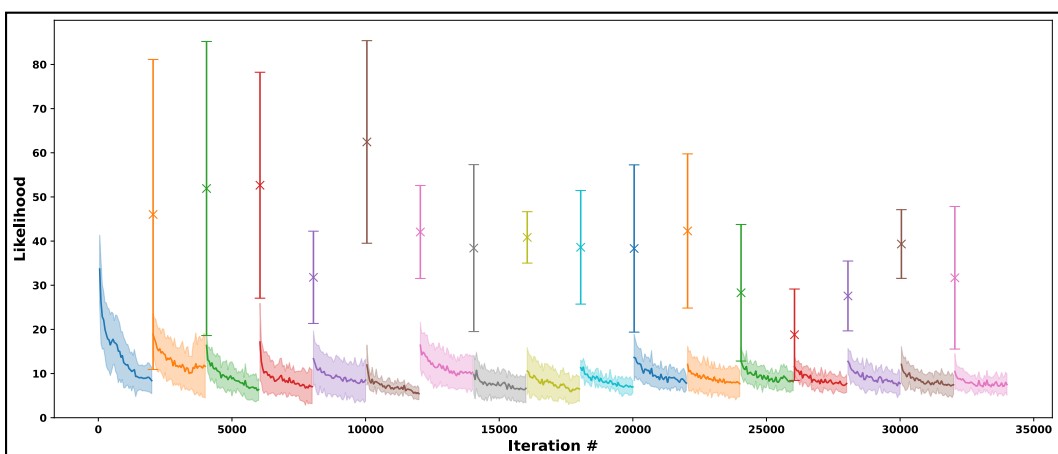

Figure 26: Progression of the feed-forward's training likelihood over tasks on **mixed-physics**, highlighting via markers the performance dip exhibited at task boundaries across the seeds. Shaded areas represent one standard deviation.

## C.8 Performance under Gradual Task Boundaries

Our task boundary detection mechanism relies on identifying performance dips during task shifts when context and query samples originate from disjoint tasks. A potential limitation of this approach arises when task shifts are gradual, resulting in a mixture of samples from the current and next tasks. In this section, we present an ablation study on increasingly mixed task boundaries, evaluating the detection accuracy of the task boundary mechanism and the overall training performance under these conditions.

To evaluate this, we tested the Task-Aware Reservoir Sampler (noting that the task boundary detection mechanism is identical in the Task-Relational setting) under a scenario where mixed batches of current and next-task samples are introduced at task boundaries. The mixed batches varied in composition, with the percentage of new-task samples incrementally decreasing from 100% to 20% of the total batch size. Following the initial mixed batch, the subsequent batch consisted entirely of new-task samples to assess whether the task boundary mechanism could correct to new the samples or if it failed due to the sliding window of likelihoods used for detection.

Fig. 27 and Fig. 28 present the 5-seed results of the boundary detection experiments on *mixed-physics* and *gravity-6* datasets, respectively. These figures show the percentage of identified task boundaries across the mixture percentages for the fixed (mixed) batch as well as the percent of additional boundaries identified on the subsequent (pure) batch. Note that the second batch percentages are additive to the mixed batch percent, so the difference between them is the performance of the correction. They additionally include the mean and standard deviation of the recorded likelihood differences. Note that, even at higher percentages of new data, the identification rate may be below 100%. This can occur in cases where technically different dynamics, based on parameter configurations, are optimized under a single cluster because the model does not find their realizations to be sufficiently distinct to justify separate clusters. This phenomenon is illustrated in Fig. 5, where two of the Hamiltonian equations (Pendulum and Mass-Spring) with unique parameter configurations collapse into 1–2 clusters each.

Table 10 and Table 11 summarize the overall performance metrics and the number of successfully identified task boundaries across varying mixture percentages. For both datasets, the total number of task boundaries differs slightly from the expected 18 (*mixed-physics*) and 12 (*bouncing-ball*) due to omitting cases where the same task appears back-to-back.

For *mixed-physics*, as the percentage of new samples decreased, the number of tasks identified during the mixed batch dropped significantly. However, the subsequent correction batch managed to capture the majority of boundaries. The mean of the mean likelihood differences steadily decreased, overlapping with the threshold hyper-parameter $\nu$ around 40%. The standard deviation of the mean likelihood difference remained stable across all percentages. For the overall performance metrics, a steady though modest decline in all metrics was observed. Despite the decline, the method still performed significantly better than the baseline gradient-based meta-learners.

In contrast, for *gravity-6*, which features harder-to-distinguish dynamics from the high-dimensional observations, performance at gradual task boundaries was notably worse. Even the correction batch succeeded in only 45% of cases as the percentage decreased. The mean likelihood difference began overlapping with $\nu$ earlier, at 60%, and its standard deviation showed a sharp drop around 40%. In the performance metrics, a significant decrease in the performance occurred across all metrics. However, we note that in this dataset still, the method performed better than the baseline gradient-based meta-learners.

The boundary detection mechanism demonstrates some resilience to gradual task boundaries, but a noticeable decline in identification success occurs as the signal produced by mismatched samples weakens. For datasets with more diverse dynamics, such as *mixed-physics*, this effect is less pronounced. However, for datasets with harder-to-distinguish dynamics, such as *bouncing-ball*, this limitation becomes more significant. Given this, auxiliary components to handle increasingly gradual task boundaries is required to extend this framework to a broader set of complex continual meta-learning settings. We note that despite the decrease in identification success, that the method still outperformed related Task-Agnostic versions. We posit that the Task-Aware setting only fully degenerates to the Task-Agnostic setting when *all* task boundaries are missed and that the inclusion of even some task-identification can still provide significant benefit to the meta-optimization. Further study of this result with additional ablation is needed, which we leave as future work.

A straightforward modification to the detection mechanism is to incorporate a recent buffer of likelihoods over steps and compare the mean of that buffer to the new likelihood. While this adjustment may not resolve issues with identifying mixed batches, it could enhance the resilience of the correction steps that follow. The current boundary detection mechanism relies solely on the likelihood from the previous step, which allows gradual task boundaries to slowly shift the distribution of likelihoods into the range of the new task without producing a significant enough difference to surpass the detection threshold $\nu$.

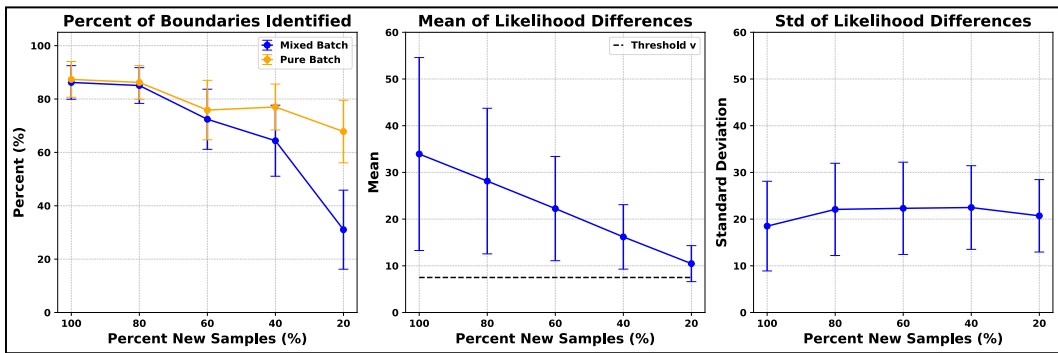

Figure 27: Gradual task boundary ablation for the Task-Aware Reservoir Sampling mechanism on **mixed-physics**. **Left)** Percent of the total boundaries identified between the mixed task boundary step and the subsequent pure batch of only new samples. **Middle)** Mean of the mean likelihood differences as calculated by boundary detection mechanism in Sec. 4.2.1. Dotted black line refers to the considered threshold hyper-parameter in experiments. **Right)** Standard deviation of the mean likelihood differences.

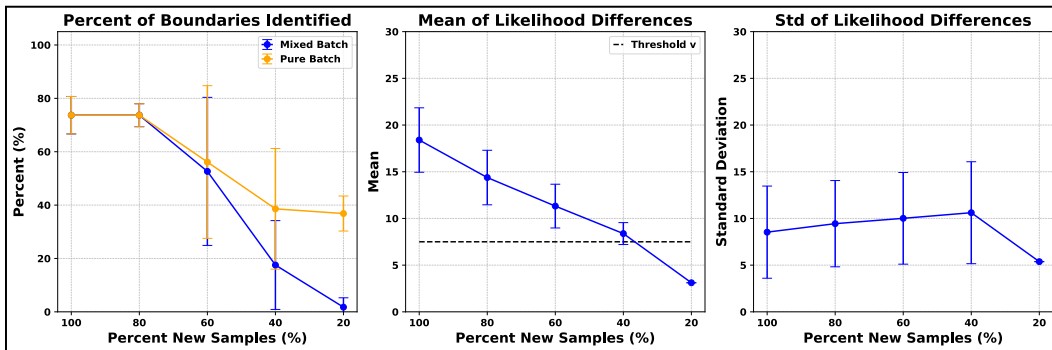

Figure 28: Gradual task boundary ablation for the Task-Aware Reservoir Sampling mechanism on **gravity-6**. **Left)** Percent of the total boundaries identified between the mixed task boundary step and the subsequent pure batch of only new samples. **Middle)** Mean of the mean likelihood differences as calculated by boundary detection mechanism in Sec. 4.2.1. Dotted black line refers to the considered threshold hyper-parameter in experiments. **Right)** Standard deviation of the mean likelihood differences.

Table 10: Performance metrics for the gradual task boundary ablation for the Task-Aware Reservoir Sampling mechanism on **mixed-physics**. % New Data represents how much of the batch at the task boundary is composed of new samples, with 20% meaning only 20% of the given batch is new samples while the remaining 80% is the current task. # Tasks Identified represents how many boundaries were successfully flagged by the boundary mechanism out of the average total boundaries.

| % New Data | # Tasks Identified | Learned | | Retained | | Interference | |
|---|---|---|---|---|---|---|---|
| | | DST↓ | MSE↓ | DST↓ | MSE↓ | DST↓ | MSE↓ |
| 100% | 16.0(0.9)/17.4 | 1.59(0.11) | 0.0143(0.0005) | 1.58(0.06) | 0.0154(0.0009) | 0.01(0.07) | -0.0012(0.0004) |
| 80% | 15.8(0.7)/17.4 | 1.72(0.22) | 0.0145(0.0006) | 1.68(0.12) | 0.0159(0.0008) | 0.04(0.11) | -0.0015(0.0002) |
| 60% | 15.0(0.9)/17.4 | 1.74(0.24) | 0.0146(0.0007) | 1.69(0.23) | 0.0155(0.0008) | 0.04(0.04) | -0.0009(0.0003) |
| 40% | 15.0(1.3)/17.4 | 1.75(0.20) | 0.0146(0.0008) | 1.69(0.16) | 0.0155(0.0009) | 0.06(0.08) | -0.0009(0.0003) |
| 20% | 13.6(1.5)/17.4 | 1.79(0.19) | 0.0149(0.0005) | 1.82(0.21) | 0.0155(0.0008) | -0.03(0.14) | -0.0006(0.0004) |

Table 11: Performance metrics for the gradual task boundary ablation for the Task-Aware Reservoir Sampling mechanism on **gravity-6**. % New Data represents how much of the batch at the task boundary is composed of new samples, with 20% meaning only 20% of the given batch is new samples while the remaining 80% is the current task. # Tasks Identified represents how many boundaries were successfully flagged by the boundary mechanism out of the average total boundaries.

| % New Data | # Tasks Identified | Learned | | Retained | | Interference | |
|---|---|---|---|---|---|---|---|
| | | DST↓ | MSE↓ | DST↓ | MSE↓ | DST↓ | MSE↓ |
| 100% | 9.0(0.6)/11.4 | 1.72(0.09) | 0.0117(0.0002) | 1.64(0.06) | 0.0118(0.0003) | 0.08(0.09) | -0.0000(0.0004) |
| 80% | 9.0(0.6)/11.4 | 1.71(0.04) | 0.0117(0.0001) | 1.66(0.08) | 0.0118(0.0002) | 0.05(0.08) | -0.0001(0.0002) |
| 60% | 7.2(3.2)/11.4 | 3.13(2.41) | 0.0135(0.0030) | 2.80(1.75) | 0.0142(0.0036) | 0.32(0.67) | -0.0008(0.0006) |
| 40% | 5.2(2.6)/11.4 | 3.61(2.26) | 0.0147(0.0030) | 2.87(1.53) | 0.0143(0.0035) | 0.75(0.81) | 0.0004(0.0012) |
| 20% | 5.4(0.5)/11.4 | 2.14(0.81) | 0.0122(0.0012) | 1.70(0.07) | 0.0115(0.0003) | 0.44(0.82) | 0.0007(0.0012) |

## D IMPLEMENTATION DETAILS

In this section, we give the specific hyper-parameters on each experiment over all models, as well as resources and considerations for each. All experiments were run on NVIDIA RTX3090 GPUs with 24GB memory in instanced cloud systems to control hardware purity. We used PyTorch 1.13.1 and scikit-learn 1.4.2 for deep learning optimization and GMM fitting, respectively.

### D.0.1 DYNAMICS SCHEMATIC VISUALIZATIONS

We present visualizations of the schematics of the considered dynamics and their governing equations in Fig. 29. In Fig. 30, we present ground truth sequence visualizations for each of the governing equations.

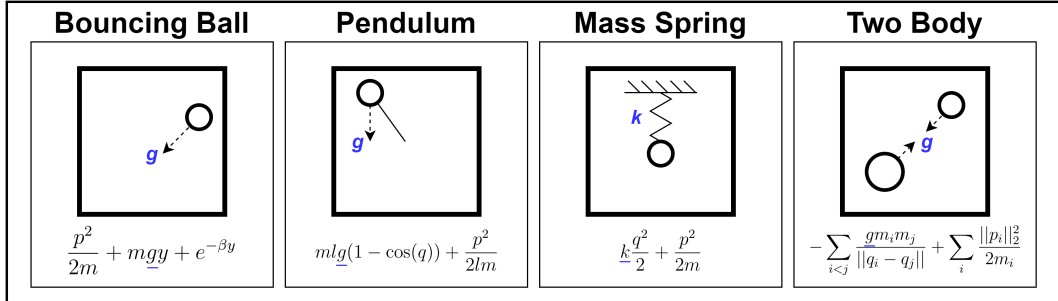

Figure 29: Schematic-based visualizations of the varying dynamics along with their underlying equations. Context variables that differentiate tasks within the same dynamic are highlighted in blue.

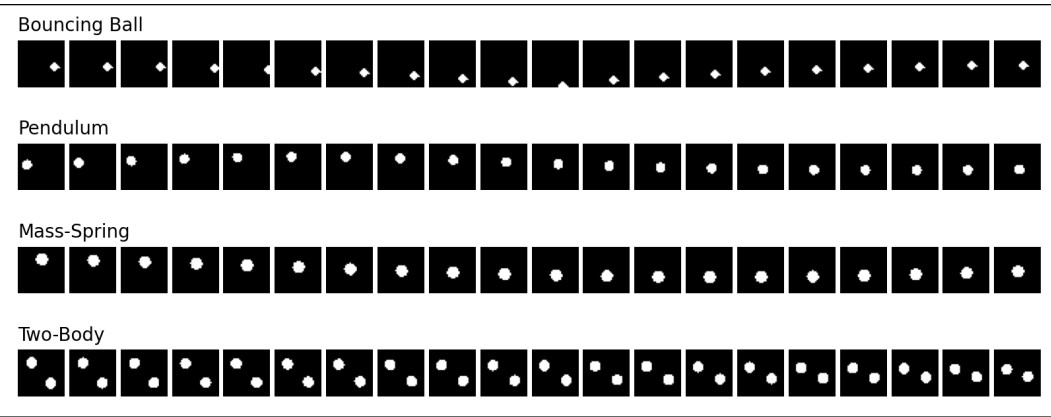

Figure 30: Image-based visualizations of the varying dynamics.

### D.0.2 DATA GENERATION DETAILS

Bouncing Ball dynamics were simulated through the PyMunk Physics Engine (www.pymunk.org), following the procedure of (Jiang et al., 2023). For **gravity-6**, we choose gravitation forces of $[60°, 120°, 180°, 240°, 300°, 360°]$ with a time simulation $\Delta t = 0.25$ and initial velocity angle limits between $[0°, 180°]$. For the 3 Bouncing Ball dynamics of **mixed-physics**, we choose gravitation forces of $[90°, 210°, 330°]$, keeping the other settings the same.

The Hamiltonian equations were simulated through the Deepmind Hamiltonian Suite (https://github.com/google-deepmind/dm_hamiltonian_dynamics_suite), following the procedures and suggested initial values described in (Botev et al., 2021). Notably, we took the red color channel from the simulations to turn them from RGB to grayscale. No friction values were considered. For the Pendulum equation, we considered three gravitational values of $[2, 3, 4]$. For the Two-Body equation, we considered three gravitational values of $[1, 2, 3]$. For the Mass-Spring equation, we considered three spring constants of $[1, 2, 3]$.

### D.0.3  EXPERIMENT IMPLEMENTATION DETAILS

All models were trained to forecast 20 timesteps using only the first 3 frames, with the exception of DKF and VRNN, which were given 8 frames for fairness. For both datasets, we set the window of *local stationarity* to 1500 iterations, where each iteration is composed of 32 active task samples. For CL, we used a reservoir size that can accommodate approximately 25% of the total data at any given time, resulting in a reservoir size of 4500 for both *gravity-6* and *mixed-physics*. Ablations on reservoir sizes and their impact are provided in Appendix C.3. The size of the $k$-shot context set was fixed at $k = 15$ for the main experiments, although we include an ablation over varying sizes in Appendix C.4. We used public implementations of DKF and VRNN for training. To ensure fairness, comparable model components to the meta-models were scaled to maintain consistent total parameter counts of 1 million trainable parameters. Shared backbone components had identical hyper-parameters, while model-specific hyper-parameters were tuned.

To test LP and RP metrics, we considered held-out testing sets of every dynamics, for both **gravity-6** and **mixed-physics**. To test LP, whenever a task boundary was detected, the task that just finished was evaluated on its testing set for its performance. At the end of the task sequence, on the resulting model, we evaluated it on every task independently and averaged their performances to get the RP metric. BTI, then, was the average difference between all task's first LP (in the event that a task re-emerged throughout the sequence) and their respective RP.

### D.0.4  ARCHITECTURE FOR META MODELS

The implementation of the considered meta-models is here: `TBD`. We use Hydra (Yadan, 2019) to handle configuration setup, in which every model's configuration file is available in the configs/ folder. The specific models, memory, and datasets to use can be changed via command-line arguments (e.g., model=metargnres memory=boundary dataset=mp). The 5 seeds considered are 1111, 2222, 3333, 4444, 5555. Detailed hyperparameter values are shown below for the feed-forward and MAML models, as well as the existing latent dynamics models. These settings are shared across memory settings and datasets.

**Feed-Forward Architecture**

- Domain Input: 20 observation timesteps of $32 \times 32$ dimensions
- Initialization Input: 3 observation timesteps of $32 \times 32$ dimensions
- Optimizer: AdamW, $5 \times 10^{-3}$ learning rate
- Gradient Norm Clipping: 5
- Transition: Recurrent Generative Network (RGN-res)
- Batch size: 32 active, 32 reservoir
- $z_t$ Latent Size: 8
- Transition Network: [SiLU(Linear(8, 64)), SiLU(Linear(64, 64)), Tanh(Linear(64, 8))]
- Domain Encoder Filters: [32, 64, 32]
- Domain Time Units: [10, 5, 1]
- Initial Encoder Filters: [32, 64, 128]
- Emission Filters: [128, 64, 32, 1]
- $z_0$ KL Beta: $\lambda_1 = 10^{-2}$
- $c$ KL Beta: $\lambda_2 = 10^{-3}$
- $c$ Cluster Loss Beta: $\lambda_2 = 10^{-3}$

**MAML Architecture**

- Domain Input: 20 observation timesteps of $32 \times 32$ dimensions
- Initialization Input: 3 observation timesteps of $32 \times 32$ dimensions
- Inner Optimizer: SGD, $5 \times 10^{-4}$ learning rate

- Outer Optimizer: AdamW, $5 \times 10^{-3}$ learning rate
- Gradient Norm Clipping: 5
- Transition: Recurrent Generative Network (RGN-res)
- Batch size: 32 active, 32 reservoir
- $z_t$ Latent Size: 8
- Transition Network: [SiLU(Linear(8, 64)), SiLU(Linear(64, 64)), Tanh(Linear(64, 8))]
- Domain Encoder Filters: [32, 64, 32]
- Domain Time Units: [10, 5, 1]
- Initial Encoder Filters: [32, 64, 128]
- Emission Filters: [128, 64, 32, 1]
- $z_0$ KL Beta: $\lambda_1 = 10^{-2}$
- $c$ KL Beta: $\lambda_2 = 10^{-3}$

## VRNN Architecture

- Input: 8 observation and 12 prediction timesteps of $32 \times 32$ dimensions
- Optimizer: AdamW, $1 \times 10^{-3}$ learning rate
- Gradient Norm Clipping: 5
- Batch size: 32 active, 32 reservoir
- Latent state dim $z_t$: 32
- Dropout probability: 0.2
- Dense X Size: 512
- Dense Z Size: 512
- Dense H(X)-Z Size: 256
- Dense H(Z)-X Size: 256
- Dense H(Z) Size: 256
- RNN Layers: 2
- RNN Dim: 128
- Beta coefficient: 1.0
- Activation: LeakyReLU(1.0)

## DKF Architecture

- Input: 8 observation and 12 prediction timesteps of $32 \times 32$ dimensions
- Optimizer: AdamW, $1 \times 10^{-3}$ learning rate
- Gradient Norm Clipping: 5
- Batch size: 32 active, 32 reservoir
- Transition: RNN Unit
- Encoder Units: [1024, 512, 256, 256]
- RNN Units: 256
- Correction Units: 256
- Transition Units: 256
- Emission Filters: [128, 64, 32, 1]

**RGNRes Architecture**

- Initialization Input: 3 observation timesteps of $32 \times 32$ dimensions
- Optimizer: AdamW, $5 \times 10^{-3}$ learning rate
- Gradient Norm Clipping: 5
- Transition: Recurrent Generative Network (RGN-res)
- Batch size: 32 active, 32 reservoir
- $z_t$ Latent Size: 32
- Transition Network: [SiLU(Linear(32, 256)), SiLU(Linear(256, 256)), Tanh(Linear(256, 32))]
- Initial Encoder Filters: [32, 64, 128]
- Emission Filters: [128, 64, 32, 1]
- $z_0$ KL Beta: $\lambda_1 = 10^{-2}$

