# OpenReview forum: "Continual Slow-and-Fast Adaptation of Latent Neural Dynamics (CoSFan): Meta-Learning What-How & When to Adapt"
_ICLR.cc/2025/Conference — ICLR 2025 Poster_

### Official Review · Reviewer_JYok · 2024-11-02

**Soundness:** 3
**Presentation:** 3
**Contribution:** 4
**Rating:** 10
**Confidence:** 4

**Summary:**

The paper introduces CoSFan, a continual meta-learning framework that adapts to changing dynamics in high-dimensional time-series data. By using a novel "what-how & when" model, it detects task shifts and quickly adjusts to new tasks, minimizing catastrophic forgetting. The claim is that CoSFan outperforms existing methods in accuracy and adaptability across non-stationary data distributions.

### Contributions
- Novel Continual Meta-Learning (CML) Framework: CoSFan, designed for continual adaptation of latent dynamics in time-series forecasting. It combines slow and fast adaptation mechanisms.

- What-How & When Framework:
  - What-How Meta-Model: Quickly adapts to specific tasks by identifying system dynamics (what) and generating task-specific parameters (how) using a feed-forward hyper-network.
  - Automatic Task Shift Detection: Identifies task boundaries to update the model for non-stationary distributions.

- Experience Replay Mechanisms:
  - Task-Aware Reservoir Sampling: Uses boundary detection to pseudo-label tasks for better context-query pairing.
  - Task-Relational Experience Replay: Clusters tasks using Bayesian Gaussian Mixture Models to handle frequent transitions and maintain rare tasks.

**Strengths:**

- Well written paper.
- Effectively combines feed-forward meta-learning with task-aware adaptation, addressing limitations in prior CML approaches reliant on gradient-based updates.
- Uses task-relational experience replay and GMM clustering to manage task transitions, a creative extension for handling non-stationary data.
- Robust empirical results validate CoSFan’s advantages, demonstrating reduced catastrophic forgetting and better adaptability than existing methods.
- Methodology is rigorous with well-defined metrics and detailed evaluation, covering key aspects of adaptation speed, memory usage, and performance retention.
- Generally clear presentation of the framework and methodology, with a structured breakdown of components ("what-how & when") aiding comprehension.
- Potential to influence further work in both CML frameworks and high-dimensional latent dynamics forecasting, with broader implications for adaptive AI in dynamic settings.
- The use of 5 seeds with average and standard deviation for all experimental results is greatly appreciated and confirms that these results are not simply a lucky seed; although the std remains very high on certain experiments.

**Weaknesses:**

- There is work to be done on the presentation of the figures (too small, need better explanations/more exhaustive captions, not clear/to many overlapping lines...)
---
- Limited to synthetic datasets (e.g., bouncing balls, Hamiltonian systems); adding real-world datasets (e.g., financial time series, climate data) would strengthen claims of applicability.
(Weakness acknowledged by the authors in limitations)
---
- The comparison with prior CML methods, especially MAML-based and task-agnostic approaches, lacks deeper discussion on specific performance trade-offs (e.g., speed vs. accuracy in adaptation).
- The added computational cost of GMM clustering and task-relational replay is not well-documented; detailing memory and processing requirements would help assess scalability.
---
- Assumes detectable task shifts and local stationarity; the model’s robustness to gradual or overlapping task shifts is not explored, limiting the generalizability to more fluid non-stationary environments.
- A quantification of how slow is to slow for the task shift to be picked up by the model would be interesting.
- Suggested improvements: include experiments with blurred task boundaries or transitions to assess flexibility in ambiguous scenarios.

**Questions:**

See weaknesses

---

> ### Author Response · Authors · 2024-11-22
> **Response to Reviewer JYok**
>
> **Presentation of results and figures**
>
> We agree that some of the figures’ presentation could be improved. We plan to continue to improve the quality of the figures.
>
> **Real world datasets**
>
> We acknowledge the need of future works to extend CosFan to real-world datasets, although the benchmark data we considered are quite representative of what are being used in the current literature of latent dynamic modeling. One challenge of identifying appropriate real-world datasets is that we are looking for time-series of high-dimensional data (e.g., image sequences) where the dynamics being learned should be in the latent space (not directly in low-dimensional data space as many available real-world time-series datasets).
>
> **Comparison on CML performance trade-offs**
>
> We did show improvements in both accuracy and speed by CosFan over MAML-based approaches, and provided further training performance comparisons to BGD-based baselines in Appendix B.2. A particularly notable performance trade-off between gradient-based methods and the feed-forward methods is that, when bi-level meta-optimization is enabled via the Task-Aware Reservoir Sampler, the gradient-methods face significant slowdowns in the face of multiple task IDs as they must sequentially process the per-task losses before aggregation to the meta-loss. We show that feed-forward methods are agnostic to the number of present dynamics within a batch and are easy to parallelize. We would like to hear from the reviewer any additional suggestions to further improve these results and discussion.
>
> **Computational cost of GMM cluster/Task-Relational replay**
>
> We appreciate the reviewer’s feedback regarding the lack of documentation on computational costs and agree that this is an important aspect to address. In response, we have added Appendix C.2, which provides analysis of memory and processing requirements across a range of reservoir sizes and increasing numbers of unique tasks. Additionally, we include a Time-to-Train comparison between the Task-Aware and Task-Relational mechanisms. Our findings show that the Gaussian Mixture Model (GMM) exhibits favorable scalability in both memory and processing time. The memory requirements for fitting the GMM on the meta-embeddings are minimal, largely due to the benefit of embedding meta-knowledge into a lower-dimensional space.
>
> **Gradual task shift experiments**
>
> We agree that exploring the stated limitation of detectable task shifts is an important aspect to include within the work. Please refer to our overall response 1 where we describe an ablation study and discussion over blurred task boundaries that was added to the work.

---

> ### Comment · Reviewer_JYok · 2024-11-25
>
> Hello,
>
> Thank you for addressing all the points and the additions to appendix C.
>
> Please make a real effort in addressing point 1 on the presentation, figures 15-19 of said appendix would be a good start.
>
> The added results on the task boundaries are very interesting, however the std of the actual data points in F.23 & 24 is very large. I am not sure they are small enough to make clear conclusions. If your pipeline is setup, running a few more seeds would be quite beneficial.
>
> Almost missed this: you have a figure mis-reference at the start of paragraph 3 of C.8. should be 23&24 if I'm not mistaken.

---

> > ### Author Response · Authors · 2024-11-27
> > **Response to Reviewer JYok**
> >
> > **Presentation of results and figures**
> >
> > We have completed a full overhaul of all the figures in the paper, excluding the data figures in Appendix D. For the revised figures, we increased font sizes, bolded all text, and converted them to a higher-quality SVG format where we could. A key improvement is in Figure 5, which has been remade for better readability, with an updated legend and caption to better distinguish between panels 5A and 5B. Additionally, Figures 15–19 have been split into two separate horizontal figures for DST and MSE, improving readability and allowing for clearer comparisons between LP and RP metrics.
> >
> > **Task boundary standard deviations**
> >
> > In case of any misunderstanding, we’d like to first clarify that the results shown in the original F23-24 (now F27-28) are the success rate of task boundary detection (left) and the average (middle) and std (right) of the mean likelihood differences as calculated by the task-boundary-detection mechanism. The performance of CoSFan in these settings were listed in Table 10-11 which does not show particularly high std. Fig27-28 now have std for the success rate (left) also. The high standard deviation in the mean likelihood differences and the detection success rate observed can be attributed to two factors. First, task boundaries between two tasks derived from the same underlying physical equation (e.g., different parameter configurations of Two-Body) may exhibit significantly smaller differences in their likelihood means compared to boundaries between tasks with more distinct dynamics. Second, as discussed in Section 5, some parameter configurations may collapse into a single cluster during the model’s optimization, effectively treating them as the same task. Consequently, certain boundaries that are technically different within these plots here as metrics may show low likelihood differences and contribute to the observed standard deviation range. Despite these factors, the majority of mean likelihood differences remains more than one standard deviation away from the threshold, which can be freely adjusted based on the variance of the expected likelihood ranges.
> >
> > **Figure mis-reference**
> >
> > This has been taken care of, thank you!

---

### Official Review · Reviewer_i4xk · 2024-11-03

**Soundness:** 3
**Presentation:** 3
**Contribution:** 3
**Rating:** 6
**Confidence:** 3

**Summary:**

This paper proposes a continual meta-learning framework (CoSFan) for forecasting high-dimensional time-series data generated by non-stationary distributions of dynamic systems. CoSFan addresses the limitations of traditional meta-learning approaches that assume stationary task distributions and known task identifiers. It proposes a feed-forward "what-how" meta-model to quickly infer the current dynamic system and adapt a latent dynamics function accordingly. Furthermore, it introduces mechanisms to detect task boundaries and employs task-aware and task-relational experience replay for continual adaptation of the model, mitigating catastrophic forgetting. Experiments on simulated physical systems demonstrate CoSFan's ability to learn and adapt to new tasks while retaining performance on previously seen ones, outperforming existing CML alternatives and standard latent dynamic models with continual extensions.

**Strengths:**

The paper is clearly written. Different components of the system and how they work together to form the final solution are clearly described.

The experiment section provides lots of insight, with studies of individual components in the framework and comparisons with other alternative solutions. These ablations are especially important for a complex system.

**Weaknesses:**

The paper opts for using hypernet as the meta-learner. Although it ablates the hypernet solution to alternatives such as MAML and Bayesian gradient descent, it lacks the comparison with a sequence leaner, such as Transformer.

**Questions:**

1. Line 215, what is s in $T_j^s$? Could you add an explaination of what s is in the text?
2. Line 216, what's z? Could you add an explaination of what z is in the text?
3. Eq. 3, what's $l$? Is it summing over $l$?
4. Line 367, typo "withou".
5. Figure 2, why is it some method task aware replay performs better than full ER?
6. Table 1, could you add a description to what number represent in the caption? Could you add the unit in the table?
7. For task-relational buffers, why is the wrong cluster assigned? Is it because of the context embedding is not well-represented? If so, is it due to the meta-learning objective? What could be done to make it better?

---

> ### Author Response · Authors · 2024-11-22
> **Response to Reviewer i4xk Pt. 1**
>
> **Comparison to additional meta-learners**
>
> We appreciate the reviewer’s suggestion that incorporating a comparison with sequence learners, such as Transformers, could enhance the robustness of the meta-learner baselines. Based on relevant work [1, 2], we have identified two promising directions to include sequence learners in our evaluation and have added an additional Related Works section dedicated to additional algorithmic priors to consider, including sequence learners. Chen et al. [1] propose a transformer-based meta-learner that combines a shared initial parameter set with available data tokens to adapt the weights, which would be conceptually similar to gradient-based meta-learner adaptation. Vladymyrov et al. [2] present a recent CML approach using a Transformer as a hyper-network to generate target network weights based on the context set. In their work, the generated weights of the previous task are used as parameter tokens for the current task’s weights, alongside active task samples. They omit the use of a replay buffer, using only the weights updated over time on active samples. While their methodology primarily focuses on image classification, adapting their approach to our latent dynamics setting is an interesting direction for future exploration. We have added this discussion in Appendix A with a reference to it in the conclusion of the main text.
>
> We refer to the overall response 2 regarding the feed-forward meta-learner and additional reasons for choosing the hyper-network specifically. Additionally, we would like to note a key advantage of our current hyper-network architecture: its parameter efficiency, as it generates the target network’s weights from a low-dimensional context embedding. Given the well-documented scaling limitations of Transformers, applying them to our setting of high-dimensional time-series may pose significant computational challenges, particularly when generating full target network parameter sets. While we have begun implementing these baselines, adapting these architectures to the forecasting setting is non-trivial. Due to the complexity of this adaptation and the rebuttal timeline, we may not be able to provide results within this review period.
>
> References:
> Chen, Yinbo, and Xiaolong Wang. "Transformers as meta-learners for implicit neural representations." European Conference on Computer Vision. Cham: Springer Nature Switzerland, 2022.
> Vladymyrov, Max, Andrey Zhmoginov, and Mark Sandler. "Continual HyperTransformer: A Meta-Learner for Continual Few-Shot Learning." Transactions on Machine Learning Research.
>
> **Explanation of s in T^s_j**
>
> $s$ refers to the context set for meta-learning. To support clarity in the text, we have added an explanation to the main text.
>
> **Line 216, variable z**
>
> $z$ refers to the latent space which the dynamics function forecasts in, and the encoder/decoders embed to and from. Section 3 Problem Formulation details what $z$ represents as well as what the equation $\mathbf{z}_t = f_\theta(\mathbf{z}_{<t}; \mathbf{c}(\mathcal{T}_j))$ represents.
>
> **Equation 3, variable l**
>
> Variable $l$ refers to the length of the observation subsequence $\mathbf{x}^q_{j,0:l}$ available to the initial state encoder derived from the full ground truth sequence $\mathbf{x}^q_{j,0:T}$. We recognize that the definition of $l$ is missing from the text and have added an explanation. Thank you for the catch.
>
> **Exact Replay metrics**
>
> Thank you for the astute observation. Based on this remark, we re-evaluated the exact-replay implementation for all baselines and identified an issue in metric reporting, which led to incorrect results. We have rerun all baselines on exact-replay and have updated figures and tables showing that, as expected, exact-replay performs the best overall in general.
>
> **Table 1 updates**
>
> We have added a description of what MAML-1 and MAML-5 represents in the table caption, as well as adding the unit of each metric to the table itself. We define metrics TTA-1 and TTA-12 within the relevant metric section, which due to space constraints of elaborating that metric, we hope is sufficiently clear.

---

> > ### Author Response · Authors · 2024-11-22
> > **Response to Reviewer i4xk Pt. 2**
> >
> > **Cluster assignment and meta-learning objective**
> >
> > As there are two potential clustering characteristics that could be referred to, and we are uncertain which one is being asked about specifically, we will define both settings before addressing each. The two characteristics are: (1) the collapse of unique parameter configurations for some Hamiltonian equations into a single general cluster, such as both Pendulum (orange) and Mass-Spring (green) in Figure 5A, and (2) the mapping of some individual embeddings to entirely incorrect tasks, as seen with certain Pendulum samples (orange) being mapped to Gravity (blue) or Two-Body (red) clusters.
> >
> > Regarding characteristic 1, this behavior is discussed in Section 5.4. These results suggest that the meta-model optimization deemed the realizations of high-dimensional observations for these individual parameter sets insufficiently diverse to justify separate clusters in the embedding space. Importantly, the performance on the Hamiltonian dynamics associated with these collapsed clusters indicates that this lack of differentiation did not negatively impact optimization.
> >
> > Regarding characteristic 2, it is true that during certain stages of optimization, some context embeddings in the reservoir may not be well-represented and can be assigned to clusters associated with different underlying tasks. We believe that additional regularization could help stabilize these context embeddings further. While the proposed cluster regularization has demonstrated improvements in stabilizing overall clusters, it does not fully address cases where individual reservoir samples are misassigned to incorrect dynamics clusters. This misassignment leads to the sample receiving an incorrect context set during experience replay, resulting in an erroneous loss signal. However, this pseudo-labeling is not permanent, as the Gaussian Mixture Model (GMM) is refit to the updated embeddings of the reservoir samples at each task boundary. Specifically, we believe that incorporating unsupervised contrastive loss terms across reservoir samples could provide a corrective signal to better align misassigned samples with their true dynamics clusters. Unfortunately, due to time constraints, we were unable to include results for this specific proposal in the rebuttal. Nonetheless, we see this as a promising avenue for future work.
> >
> > **Other minor edits**
> >
> > All corrected as suggested, thank you!

---

> > > ### Comment · Reviewer_i4xk · 2024-11-27
> > >
> > > Thank you for the detailed reply, as well as the corrections/clarifications to the paper.
> > >
> > > Apart from using the Transformer with a clear meta-learning approach, you could also consider using it with an online learning approach such as in [1] which works well with a replay.
> > >
> > > Overall, I am happy with the reply and would like to maintain the score.
> > >
> > > [1] J. Bornschein et al. Transformers for Supervised Online Continual Learning.

---

### Official Review · Reviewer_U6Tv · 2024-11-04

**Soundness:** 3
**Presentation:** 3
**Contribution:** 3
**Rating:** 5
**Confidence:** 3

**Summary:**

This paper proposed a new method for continual meta-learning (CML). Different from previous methods (e.g. MAML variants), the proposed method uses a single feed-forward network. Using the contex encoder and hyper network, the support samples are encoded to context vector, and the context vector is used to produce the parameters for encoded query samples using the hyper network. Furthermore, the authors also proposed task-aware reservoir sampling approach by using the gaussian mixture model to identify the tasks. In the experiment section, the proposed methods outperforms the gradient-based meta learners, and also show the effectiveness of using task-aware sampling strategy.

**Strengths:**

Strengths

1. Different from the gradient-based methods which needs a number of gradient steps to adapt, the proposed methods can adapt to novel tasks using a sinlge feed-forward network.

**Weaknesses:**

Weaknesses

1. I think the motivation behind each component is weak. For example, for adapting to the query samples, the CosFan uses the hyper network to produce the parameters. However, why we should use the hyper network framework in this feed-forward network? Isn't it possible to use other trainalbe embedding networks for the query samples?

2. The mechanism for detecting the task boundary is not novel. The detection mechanism simply comes from the methods in [1]. However, though the authors show that the advantage of CosFan over other baselines lies on the ability for detecting the task boundary without any assumptions on the task identifiers, I think other baselines can adopt the task boundary detection mechanism used in this paper with simple modification. Therefore, I don't think the ability on detecting the task boundary in CosFan is advantage compared to other methods

3. The overall notation is confusing. For example, in line 217~220, what is the measing of superscript s in $T_j$? Is it support set? I think it is not clear. Furthermore, in the definition of $T_j$, the internal sequences are not dependent on $j$. I think the authors should clarify all the notations to prevent the confusion.


[1] Caccia et. al., Online Fast Adaptation and Knowledge Accumulation (osaka): A New Approach to Continual Learning, NeurIPS, 2020

**Questions:**

Already mentioned in weaknesses section

---

> ### Author Response · Authors · 2024-11-22
> **Response to Reviewer U6Tv**
>
> **Component motivation and trainable embedding networks**
>
> Please refer to our overall response 2 for clarification on the feed-forward methodology and experiments on comparing alternative conditioning mechanisms, such as trainable embedding networks.
>
> **Task boundary detection novelty**
>
> Please refer to our overall response 3 for clarification on the novelty of the task boundary detection mechanism and the primary contribution of the work.
>
> **Clarification of notations**
>
> Thank you for bringing up confusions in the notation. Indeed the superscript s refers to the context set for meta-learning. We have done a passthrough notation and have cleaned up notation throughout the paper, including making the internal sequences of T_j dependent on j.

---

> > ### Author Response · Authors · 2024-11-27
> > **Reminder to Reviewer U6Tv**
> >
> > Dear Reviewer U6Tv
> >
> > Thank you again for your time and efforts in reviewing our paper. As the deadline for discussion is approaching, we hope that you had time to review our revised manuscript and responses. We would like to follow up to see if you have additional comments for further discussion.
> >
> > Best regards,
> >
> > Authors

---

> > > ### Author Response · Authors · 2024-12-02
> > > **Reminder to Reviewer U6Tv**
> > >
> > > Dear Reviewer U6Tv
> > >
> > > We truly appreciate your time and effort in reviewing our submission. The rebuttal discussion period comes to a close soon, and we would like to confirm whether our responses, added ablations, and manuscript updates have effectively addressed your questions and concerns.
> > >
> > > Thank you!
> > >
> > > Authors

---

### Official Review · Reviewer_GEod · 2024-11-05

**Soundness:** 2
**Presentation:** 3
**Contribution:** 2
**Rating:** 6
**Confidence:** 3

**Summary:**

This paper introduces CoSFan, a method for meta continual learning that the paper evaluates on sequences of dynamical systems tasks. The approach consists of a hyper network based approach where the parameters of the current task being adapted to are generated using a forward pass of the hypernetwork on a context vector created using the average encoding of the samples in a context set. The generated parameters are used in a dynamics model that predicts the latent of the next element in the sequence and is then decoded. The meta-parameters and the encoder/decoder are optimized using the MSE between predicted and ground truth query sequences. The paper also explores how to detect task boundaries to manage/balance the number of samples in a replay buffer being used to rehearse on previous tasks in the sequence. It proposes two different mechanisms: one where it looks at any spike in loss as a task change, and one where it use a gaussian mixture model to cluster examples and detect whether any new clusters have formed.

The paper evaluates on a series of image based dynamical systems tasks where the physics/gravity constants are changed in each task, and the model must predict the next state. The paper shows an improvement in a slight improvement in learning performance over other meta learning approaches, but shows a clear improvement in forgetting over other approaches when using both their meta learning approach and the task detection based replay.

**Strengths:**

- Figure 5 shows a very clear use case for the task relational replay over even a task aware (and presumably task agnostic) replay.
- Figure 4 shows that the meta learning approach proposed by the paper helps mitigate forgetting when task detection is used.
- The method proposed can adapt to new sequences/new samples much faster than gradient based adaptation approaches, since they don’t need to take any gradient steps.

**Weaknesses:**

- The way Figures 2 and 4 are presented is a bit confusing. For one, they use different metrics, and it’s unclear what the insight from using one over the other is. For figure 2, the red bar is the only meta-learned method, so the other 3 methods were presumably evaluated in a different setting. Were they given the “context” information? It seems the more direct comparison with baselines is in Figure 4.
- It’s unclear if the hypernetwork based approach can scale better than an adaptation based approach when the number of heterogeneous tasks increases.
- As the authors do mention, the task shift detection seems to rely on abrupt task shifts, and would likely fail on gradual task shifts. In the case where the task shift detection fails, and the default is to task agnostic reservoir sampling, this method seems to do slightly worse than other meta learning based approaches.

**Questions:**

- For the experiments depicted in Figures 2 and 4, it seems each task is trained for the same number of samples. In this setting, why is there a big difference in the retained performance of specifically the meta learned methods between task agnostic and task aware replay? Wouldn’t the number of samples of each task still be approximately the same given that the replay size per task is still in the hundreds? Is the difference of a few samples really making that big of a difference?
- How does the hypernetwork based method scale with the number of heterogeneous tasks?
- I am a bit unclear about the setting. Is there a different context set sampled for each query example? Or is it just that the previous k sequences is treated as the context set?
- It would be interesting to see an adaptation based baseline which was also given the context vector as input.
- Do you have a version of Figure 3 comparing the Task aware replay with the task relational replay?

---

> ### Author Response · Authors · 2024-11-22
> **Response to Reviewer GEod Pt. 1**
>
> **Clarification of metrics and intended results in Figures 2 and 4**
>
> We used both MSE and DST across all experiments as MSE measures averaged pixel-level accuracy while DST the average Euclidean distance between the predicted object’s location and its ground truth. Due to space restrictions we are not able to report complete results from all metrics in all experiments in the main text – we thus decided to use DST as an example in Fig 2 whereas MSE in Fig 4; complete results on both metrics are in Appendix.
>
> Results in Figure 2 and Figure 4 are intended to demonstrate different points. Results in Figure 2 (Section 5.2) are to show that both continual and meta methods are required simultaneously to achieve good performance. The non-meta models use CL strategies (except in naive learning) without context data to demonstrate the benefit of meta-learning (i.e., knowing how to adapt) in this setting. Because latent dynamic forecasting has not been studied in this setting of non-stationary and heterogeneous dynamics distributions, we felt that it is important to first demonstrate that both continual and meta components are essential in this setting. Once this is established in Section 5.2 / Figure 2, Figure 4 (and Section 5.3) then shows how the proposed continual and meta strategies improve over existing CML works (the baselines).
>
> **Task shift experiments**
>
> We agree with the reviewer’s observation that the method performs slightly worse than MAML-based meta-learners in the task-agnostic setting. Please refer to our overall response 1 for an additional ablation study we performed on gradual task shift experiments and additional justification as to the advantages of Task-Aware replay over Task-Agnostic replay.
>
> **Differences between Task-Agnostic and Task-Aware methods**
>
> Indeed each task is trained with the same number of samples, and both Task-Agnostic and Task-Aware Replays maintain an approximately equivalent sample distribution across tasks during training, based on similar usage of the Reservoir Sampling algorithm.
>
> Please refer to our overall response 3 for our clarification on the key difference between the Task-Aware methodology over the Task-Agnostic approach.
>
> **Hypernetwork scaling**
>
> There is no required scaling with respect to the number of tasks in either the parameter or input size of the feed-forward hyper-network approach. It is more a matter of representational capacity of the network with respect to the complexity of the underlying task diversity and distribution. We argue that the algorithmic prior of this learned function transformation provides better scaling than gradient-based techniques, which have to share manifold capacity from an initial static point across the tasks. Computationally, the feed-forward adaptation approach benefits from being inherently agnostic to the number of heterogeneous tasks and efficiently parallelizing the adaptive forward pass across context sets. In contrast, gradient-based meta-learners require costly per-task test-time fine-tuning. This efficiency advantage is demonstrated in the adaptation efficiency comparison presented in Table 1.
>
> **Clarification on context-query pairing**
>
> For the task which is actively streaming in T_j, the previous k sequences are treated as the context set for the active samples. However, when we sample from the reservoir to get past-task query samples for experience replay, we additionally sample a relevant context set from the reservoir based on their assigned pseudo-labels (obtained by our two task identification strategies)

---

> > ### Author Response · Authors · 2024-11-22
> > **Response to Reviewer GEod Pt. 2**
> >
> > **Adaptation-based baselines**
> >
> > We would appreciate it if the reviewer could clarify what “an adaptation based approach” is intended to refer to, as both the proposed feedforward meta-learner and the MAML-based approaches we used as one of the baselines are adaptation based approaches (the latent dynamic functions are learned to be adapted to context data). Based on our understanding, the reviewer may be referring to an alternative baseline where, after having derived the context variable $\mathbf{c}(\mathcal{T}_j)$ from the context set, we provide the context variable as additional input to condition/adapt the latent dynamics function. This formulation of feed-forward adaptation aligns with the approach considered in prior work [1].
> >
> > Please refer to the overall response 2 for our clarification on the choice of the hyper-network as well as experiments including this alternative baseline.
> >
> > We kindly request clarification from the reviewer to confirm if this interpretation aligns with the intended meaning of “adaptation-based baseline,” so that we can further address any misunderstandings in our response.
> > .
> > [1] Jiang, Xiajun, et al. "Sequential latent variable models for few-shot high-dimensional time-series forecasting." The Eleventh International Conference on Learning Representations. 2023.
> >
> > **Figure to compare task-aware and task-relational replays**
> >
> > We appreciate the suggestion for adding this visualization. In the Appendix B.3 dedicated to Task-Aware vs. Task-Relational comparisons, we have added two additional figures comparing them on both mixed-physics and gravity-6.

---

> > > ### Comment · Reviewer_GEod · 2024-11-25
> > >
> > > I still am not convinced about the scaling of hypernetworks. As you mentioned, the representational capacity of the hypernetwork is limited, and as you scale the number of heterogeneous tasks, you might hit that limit. Meanwhile, for a gradient adaptation type baseline, it isn't necessary to store all the information about a task in the network, as you are allowed to adapt.
> > >
> > > Overall, though I am satisfied with the rest of the responses and am raising my score.

---

> > > > ### Author Response · Authors · 2024-11-27
> > > > **Response to Reviewer GEod**
> > > >
> > > > **Hypernetwork scaling**
> > > >
> > > > Thank you for raising the point in terms of the adaptation capacity between a gradient adaptation baseline and a feedforward model-based adaptation method. While the latter offers the advantage of removing the need of gradient-based adaptation at test time, it is true that the feedforward model may encounter capacity limits as the number (or complexity) of the heterogeneous tasks increases. We have not yet observed this limit in our experiments so far, as demonstrated in our results, but we will take a deeper dive into this issue both theoretically and experimentally in our future work.

---

### Author Response · Authors · 2024-11-22
**Summary of major additions**

We thank the reviewers for their constructive feedback. While detailed responses have been provided to individual reviewers, we highlight the main additions and clarifications made to address the key concerns raised here. Updated sections within the manuscript are highlighted in blue.

**1. Ablation Study on Boundary Detection with Gradual Task Shifts (Reviewers GEod and JYok).**

In response to suggested improvements regarding the limitation of detectable task shifts, we have added Appendix C.8, which evaluates the boundary detection mechanism under increasingly blurred task boundaries on the mixed-physics and gravity-6 datasets. This study evaluates both the success of boundary identification and its impact on the overall forecasting performance. Our results show that boundary identification rates remain stable up to 40–60% overlap between current and next task data, after which they drop significantly. Forecasting performance varies by dataset: for mixed-physics with more heterogeneous dynamics, performance decreases modestly with increasing overlap but remains relatively stable; for gravity-6 with a lower level of heterogeneity among dynamics , performance declines more significantly when overlap exceeds 60%. Despite this, the Task-Aware setting still consistently outperforms the Task-Agnostic setting on both datasets in all overlap levels. We thank the reviewers for suggesting this set of new experiments, which demonstrates a notable level of resilience of CoSFan to gradual boundaries and provides additional concrete evidence for the advantages of the proposed Task-Aware strategies to the state-of-the-art Take-Agnostic strategies used in CML. We attribute this to the idea that Task-Aware methods only fully degenerate to Task-Agnostic methods when all task boundaries are missed, with partial task identification still providing meaningful benefits for meta-optimization. In Appendix C.8, we also provide analyses of failure cases, linking them to task likelihood variance and boundary swap rates, and propose potential modifications to improve robustness under these scenarios. In addition, we have added a relevant pointer in the Conclusion of the main text towards the results of this ablation.

**2.Clarification of Adaptation-Based Methodology and Comparison of Conditioning Mechanisms (Reviewers GEod and U6Tv).**

In response to reviewer GEod, We would like to clarify that the proposed method is fundamentally adaptation-based, utilizing feed-forward models to adapt the latent dynamics function with meta-knowledge from the context set. In response to reviewers U6Tv and GEod, we would like to further clarify that our main contribution on the meta-learner is to demonstrate the benefit of feed-forward meta-learners over widely-used MAML-based meta-learners in this problem: hyper-network based (multiplicative) adaptation is a specific design choice we used for this feed-forward meta-learner; it is neither the only choice nor the key innovation of CoSFan. To demonstrate the generality of CoSFan beyond this specific choice, we added an alternative approach using an embedding conditioning based (additive) adaptation, where the derived context variable is concatenated with the latent state of the dynamics function, aligning with prior meta-learning work [1]. Results are added to Appendix C.1 on mixed-physics and gravity-6 datasets, showing that CoSFan’s performance is not affected by this design choice (embedding mechanism performs comparably in the considered datasets). In the original manuscript, hyper-network architecture was chosen for its demonstrated suitability across diverse domains and its ability to model interactions that subsume those provided by embedding-based (or additive-based) conditioning [2], providing additional representation capacity without compromising optimization stability..
.
[1] Jiang, Xiajun, et al. "Sequential latent variable models for few-shot high-dimensional time-series forecasting." The Eleventh International Conference on Learning Representations. 2023.
[2] Jayakumar, Siddhant M., et al. "Multiplicative interactions and where to find them." International conference on learning representations. 2020.

---

> ### Author Response · Authors · 2024-11-22
> **Summary of major additions Pt. 2**
>
> **3. Clarifying on Contribution Related to Task Identification and Bi-Level Optimization (Reviewer GEod).**
>
> We would like to clarify the benefit of the Task-Aware methodology over the Task-Agnostic approach as it underpins a key contribution of the paper: our main innovation is not on the boundary detection mechanism, but the task identification strategies used to enable per-task context-query pairing to enable bi-level meta-optimization that is not achieved in existing CML approaches. The Task-Agnostic Replay mechanism does not leverage task information, relying instead on the approximate equivalence between meta-learning and continual learning objectives in aligning the current task’s gradient with the average gradient of previous tasks. This involves adapting only on the current task’s context set and using the resulting parameter set on the past tasks’ samples to get the meta-loss. While sufficient for image classification and low-dimensional regression problems in prior work, we found this approach inadequate for latent dynamics forecasting. In contrast, the Task-Aware Replay mechanism incorporates the estimated task ID via the boundary detection mechanism, enabling traditional per-task context-query pairing for reservoir samples. This allows each task to adapt to its relevant context set sampled from the reservoir, with per-task losses aggregated to form the meta-loss. The enables meta-learning to be done in a continual fashion without compromising full bi-level meta-optimization, which we believe was the cause of the significant performance gain in the proposed task-aware over task-agnostic settings across both the gradient-based and our considered feed-forward meta-learners.
>
> **4. Documentation of Computational Costs (Reviewer JYok).**
>
> To address suggestions regarding the computation cost of the Task-Relational Experience Replay, Appendix C.2 has been added to document the computational costs of Gaussian Mixture Model (GMM) clustering and task-relational replay. We analyze memory and processing requirements across varying reservoir sizes and task numbers and include a Time-to-Train comparison between Task-Aware and Task-Relational mechanisms. Our findings show that the GMM scales favorably in both memory and processing time, with minimal memory requirements due to the low-dimensional embedding of meta-knowledge.

---

> > ### Author Response · Authors · 2024-11-27
> > **Summary of major additions Pt. 3**
> >
> > **5. Presentation of results and figures.**
> >
> > We have completed a full overhaul of all the figures in the paper, excluding the data figures in Appendix D. For the revised figures, we increased font sizes, bolded all text, and converted them to a higher-quality SVG format where we could. A key improvement is in Figure 5, which has been remade for better readability, with an updated legend and caption to better distinguish between panels 5A and 5B. Additionally, Figures 15–19 have been split into two separate horizontal figures for DST and MSE, improving readability and allowing for clearer comparisons between LP and RP metrics.

---

### Meta-Review · Area_Chair_uowX · 2024-12-24

**Metareview:**

Reviewers remarked positively about the proposed approach, agreeing that the method is effective in mitigating forgetting when task identification is successful, remarking on the clear use case for the task relational replay and identifying the faster adaptation time as a major advantage over gradient-based methods. Apart from several smaller items mentioned by reviewer JYok, the submission was generally perceived to be clearly written and considered strong in its empirical methodology.

On the downside, questions remain about the scaling of the proposed hypernetwork-based approach with the number of heterogeneous tasks and the applicability of the setup to problems with gradual shifting instead of abrupt task changes (a condition in which the task detection mechanism will struggle, as confirmed by the authors). Results could be further improved by moving beyond synthetic datasets.

On balance, this submission is above the acceptance threshold, with one reviewer providing very strong feedback in favour of the submission. This submission should thus be accepted for publication at ICLR.

**Additional Comments On Reviewer Discussion:**

Healthy Reviewer Discussion apart from Reviewer U6Tv, who did not respond to the rebuttal. As a result, I slightly down-weighted their criticism in my overall assessment.

---

### Decision · Program_Chairs · 2025-01-22

Accept (Poster)